# Coordinated multiplexing of information about separate objects in visual cortex

Na Young Jun[1,2,3]*, Douglas A Ruff[4,5], Lily E Kramer[4,5], Brittany Bowes[4,5], Surya T Tokdar[6], Marlene R Cohen[4,5], Jennifer M Groh[1,2,3,7,8,9]*

[1]Department of Neurobiology, Duke University, Durham, United States; [2]Center for Cognitive Neuroscience, Duke University, Durham, United States; [3]Duke Institute for Brain Sciences, Durham, United States; [4]Department of Neuroscience, University of Pittsburgh, Pittsburgh, United States; [5]Center for the Neural Basis of Cognition, University of Pittsburgh, Pittsburgh, United States; [6]Department of Statistical Science, Duke University, Durham, United States; [7]Department of Psychology and Neuroscience, Duke University, Durham, United States; [8]Department of Biomedical Engineering, Duke University, Durham, United States; [9]Department of Computer Science, Duke University, Durham, United States

*For correspondence:
nayoung.jun@duke.edu (NYJ);
jmgroh@duke.edu (JMG)

## Abstract

Sensory receptive fields are large enough that they can contain more than one perceptible stimulus. How, then, can the brain encode information about *each* of the stimuli that may be present at a given moment? We recently showed that when more than one stimulus is present, single neurons can fluctuate between coding one vs. the other(s) across some time period, suggesting a form of neural multiplexing of different stimuli (Caruso et al., 2018). Here, we investigate (a) whether such coding fluctuations occur in early visual cortical areas; (b) how coding fluctuations are coordinated across the neural population; and (c) how coordinated coding fluctuations depend on the parsing of stimuli into separate vs. fused objects. We found coding fluctuations do occur in macaque V1 but only when the two stimuli form separate objects. Such separate objects evoked a novel pattern of V1 spike count ('noise') correlations involving distinct distributions of positive and negative values. This bimodal correlation pattern was most pronounced among pairs of neurons showing the strongest evidence for coding fluctuations or multiplexing. Whether a given pair of neurons exhibited positive or negative correlations depended on whether the two neurons both responded better to the same object or had different object preferences. Distinct distributions of spike count correlations based on stimulus preferences were also seen in V4 for separate objects but not when two stimuli fused to form one object. These findings suggest multiple objects evoke different response dynamics than those evoked by single stimuli, lending support to the multiplexing hypothesis and suggesting a means by which information about multiple objects can be preserved despite the apparent coarseness of sensory coding.

## Editor's evaluation

The authors report that neurons in V1 and V4 multiplex information of simultaneously presented objects. A combination of multi-single unit recordings, statistical modelling of neuronal responses and neuronal correlations analyses argues in favor of their claims. Pairs of neurons having similar object preferences tended to be positively correlated when both objects were presented, while pairs of neurons having different object preferences tended to be negatively correlated and these patterns and others suggest that information about the two objects is multiplexed in time. These results are of broad interest to the field, as they shed new light on the "binding" problem and highlight the importance of underexplored features of cortical activity for neural coding.

## Introduction

Coarse population coding has been widely explored in motor systems, where neurons show broad activity profiles and are thought to 'vote' for the movement typically associated with their peak activity (e.g., *Georgopoulos et al., 1986*; *Lee et al., 1988*). However, individual motor systems only generate one movement at a time. Such a coarse coding/population voting scheme cannot work in sensory systems where there are generally many stimuli to be represented rather than a single (e.g., arm or eye) movement to be specified. It has been assumed that sensory receptive fields are small enough that coarse coding does not apply, but this seems questionable. For example, the letters on the page you are reading now are probably <0.25° apart, but foveal V1 receptive fields are approximately 0.5–2° in diameter (*Dow et al., 1981*; *Alonso and Chen, 2009*; *Xing et al., 2009*; *Dubey and Ray, 2016*; *Keliris et al., 2019*). Receptive fields get even larger at later stages along the visual processing stream (e.g., *Alonso and Chen, 2009*). In the auditory system, mammalian neurons may be responsive to nearly any location in space (e.g., *Woods et al., 2001*; *Groh et al., 2003*; *McAlpine and Grothe, 2003*; *Werner-Reiss and Groh, 2008*; *Grothe et al., 2010*; *Higgins et al., 2010*) and even frequency tuning is broad at conversational sound levels (*Bulkin and Groh, 2011*; *Willett and Groh, 2022*). Such breadth of tuning means that there can be overlap in the population of neurons activated by individual stimuli, making it unclear how information about multiple objects is preserved.

Logic suggests that information about each distinct stimulus must be segregated within the neural code in some fashion, either into exclusive neural subpopulations, different epochs of time, or some combination of both. We have recently presented the hypothesis that the nervous system might employ a form of neural turn-taking (time division multiplexing) in which individual neurons fluctuate between responding to each of the items in or near their receptive fields across various epochs of time (*Caruso et al., 2018*; *Mohl et al., 2020*). Such a coding scheme could preserve information about each stimulus across time and/or across neural subpopulations.

This theory raises three key open questions. First, is multiplexing a general phenomenon that occurs across a range of different brain areas? Our original study tested one subcortical auditory area (the inferior colliculus) and one extrastriate visual cortical area (area MF of the inferotemporal [IT] face patch system). More areas need to be tested to understand how such a coding scheme might operate. Second, in brain areas that exhibit such coding fluctuations, do neurons fluctuate together, and if so, how? Pairs of neurons might show positive, negative, or no correlations with each other. The pattern of such correlations across the population can reveal whether the population as a whole retains information about both stimuli and whether there is bias favoring one stimulus over another.

Third, does the pattern of fluctuations depend on the parsing of the scene into separate objects? Individual stimuli can fuse into one object or be perceived as distinct from each other. Stimuli that segregate into separate objects may be more likely to be associated with fluctuations in neural activity and their attendant correlations across neurons (*Milner, 1974*; *Gray and Singer, 1989*; *Von Der Malsburg, 1994*; *Singer and Gray, 1995*; *Gray, 1999*) (but see *Palanca and DeAngelis, 2005*), whereas stimuli that fuse into a single distinct object may cause activity patterns that are akin to those observed when only one stimulus is present. Such a pattern would specifically implicate activity fluctuations in playing a role in the perceptual process of object segregation.

To address these questions, we turned to the primary visual cortex (V1). V1 allows for a strong test of these hypotheses since V1 neurons have comparatively small receptive fields and are therefore less subject to the multiple-stimulus-overlap problem than the more broadly tuned areas such as the inferior colliculus or inferotemporal (IT) cortex that were assessed in our previous report (*Caruso et al., 2018*). Even though V1 neurons themselves have comparatively small receptive fields, V1 contributes to processing in higher cortical areas where spatial tuning is coarser. V1 could therefore also exhibit fluctuating activity patterns so as to facilitate preservation of information about multiple stimuli at higher stages.

We evaluated activity in V1 while monkeys viewed either individual stimuli (gratings) or two different types of combined stimuli (superimposed vs. adjacent gratings). When the two gratings were superimposed, they presumably appeared as one fused object, or plaid (*Adelson and Movshon, 1982*; *Rodman and Albright, 1989*; *Heeger et al., 1996*; *Busse et al., 2009*; *Lima et al., 2010*), whereas when they were adjacent they appeared as two distinct objects. We found evidence for coding fluctuations when two gratings were present at separate locations (two objects) but not when the gratings were superimposed at the same location and appeared as one fused object. We then evaluated the

**Table 1.** Summary of included data.

Analyses were conducted on 'triplets,' consisting of a combination of A, B, and AB conditions. If the spikes evoked by the A and B stimuli failed to follow Poisson distributions with substantially separated means, the triplet was excluded from analysis. This table shows the numbers of triplets that survived these exclusion criteria for each brain area and type of stimulus condition (last column), as well as the numbers of monkeys, distinct units, and sessions that they were derived from (columns 6–9).

| 1. Stimuli | 2. Brain area | 3. Task | 4. Monkeys | 5. Available sessions | 6. Sessions for which at least one triplet was included | 7. Available units | 8. Units for which at least one triplet was included | 9. Triplets passing exclusion criteria for analysis |
|---|---|---|---|---|---|---|---|---|
| Adjacent | V1 | Attention | ST, BR | 16 | 16 | 1604 | 935 | 1389 |
|  | V4 | Fixation | BA, HO | 17 | 17 | 991 | 274 | 456 |
| Superimposed | V1 | Fixation | ST, BR | 25 | 23 | 2304 | 770 | 1686 |
|  | V4 | Fixation | JD, SY | 21 | 21 | 1744 | 817 | 1529 |

degree and sign of the spike count correlations (commonly referred to as 'noise' correlations; ***Cohen and Kohn, 2011***) observed between pairs of simultaneously recorded units in response to presentations of particular stimulus conditions. We found that the pattern of correlations varied dramatically depending on whether the stimuli were presented either individually or superimposed (single stimuli or one fused object) vs. when they were presented side-by-side (two separate objects). In the two-object case, the distribution of spike count correlations was markedly different from previous reports involving individual stimuli (***Table 1***, ***Cohen and Kohn, 2011***), and encompassed a range spanning many negative correlations in addition to positive ones. Whether the correlations tended to be positive vs. negative depended on whether the two neurons in the pair preferred the same stimulus (median correlation + 0.25) or preferred different stimuli (median correlation –0.05). The distribution of spike count correlation values was even more widely spread among pairs of neurons that showed demonstrably fluctuating activity across stimulus presentations (same preference: +0.49 and different preference: –0.14). In contrast, in the single stimuli and fused object (superimposed gratings) cases, positive correlations predominated (single stimuli: median value 0.15–0.19; fused object: median value +0.15). Distinct tuning-preference-related distributions of spike count correlations for adjacent stimuli but not for superimposed/fused stimuli were also seen in a smaller additional dataset in V4.

Overall, this pattern of results is consistent with the possibility that when two visual objects are presented in close proximity, a subpopulation of visual cortical neurons fluctuates in a coordinated fashion, generally retaining information about segregated objects and suggesting an account for why they can be perceived at once.

## Results

### General experimental design

The activity of neurons in visual cortex was recorded in three experimental designs in six monkeys (N = 2 per experiment per brain area), using chronically implanted multielectrode arrays (***Figure 1a***, ***Table 1***). In the 'superimposed' dataset, the activity of neurons in V1 and V4 was recorded while monkeys passively fixated (for details, see ***Ruff et al., 2016***). In the 'adjacent' datasets, the activity of V1 and V4 neurons was recorded while monkeys either passively fixated (V4 recordings) or fixated while performing an orientation change discrimination task involving either one of these stimuli or a third stimulus presented in the ipsilateral hemifield (V1) (for details, see ***Ruff and Cohen, 2016***). In the 'superimposed' dataset, the gratings were large, spanning the receptive fields of the recorded neurons, and were presented either individually or in combinations of two orthogonal gratings at a consistent location on every trial (***Figure 1c***). When the two gratings were presented, they superimposed and formed one fused 'plaid' object. In the 'adjacent' datasets, the stimuli were smaller Gabor patches (V1, V4, ***Figure 1d and e***) or natural images (V4, ***Figure 1e***, stimuli from ***Long et al., 2018***) and were presented either individually or adjacent to one another as two separate objects. Together they spanned the receptive fields of the V1 or V4 neurons being recorded in a fashion similar to the 'superimposed' experiment. For data collected during performance of the attention task (V1), we

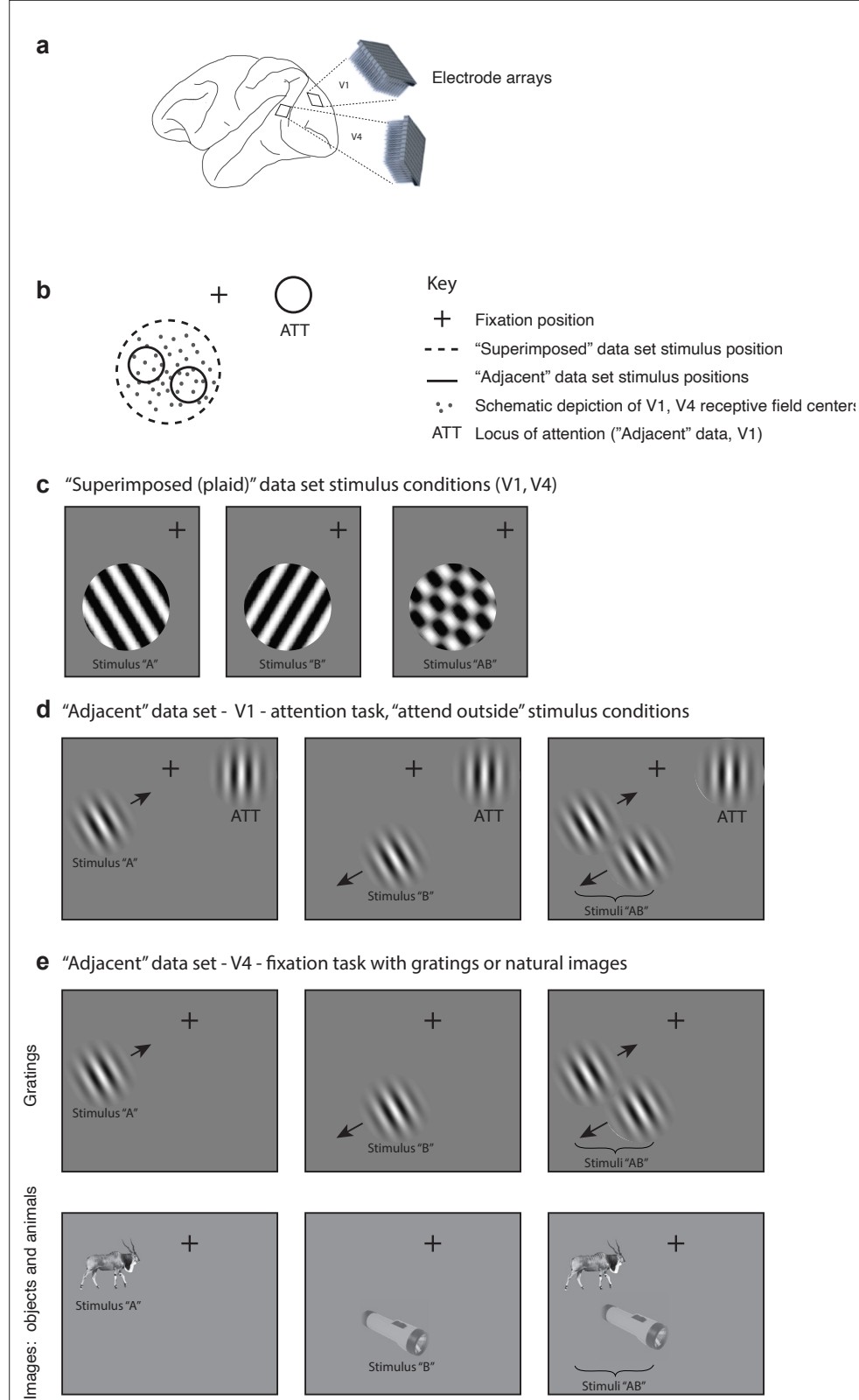

**Figure 1.** Experimental design. (**a**) Multiunit activity was recorded in V1 and V4 using chronically implanted 10 × 10 or 6 × 8 electrode arrays in six monkeys (see 'Methods'). (**b**) In both 'superimposed' and 'adjacent' datasets, the stimuli were positioned to overlap with ('adjacent' dataset) or completely span ('superimposed' dataset) the centers of the receptive fields of the recorded neurons. (**c**) In the 'superimposed' dataset, gratings

*Figure 1 continued on next page*

*Figure 1 continued*

were presented either individually or in combination at a consistent location and were large enough to cover the V1 and V4 receptive fields (stimulus diameter range: 2.5–7°). The combined gratings appeared as a plaid (rightmost panel). Monkeys maintained fixation throughout stimulus presentation and performed no other task. (**d**) In the V1 'adjacent' dataset, Gabor patches were smaller (typically ~1°, see *Figure 1—figure supplement 1*) and were presented individually or side-by-side roughly covering the region of the V1 receptive fields. Monkeys maintained fixation while performing an orientation change detection task. The data analyzed in this study involved trials in which the monkeys were attending a third Gabor patch located in the ipsilateral hemifield to perform the orientation change detection. (**e**) In the V4 'adjacent' dataset, the stimuli consisted of either Gabor patches or natural image stimuli, and monkeys performed a fixation task. Incorrectly performed trials and stimulus presentations during which we detected microsaccades were excluded from all analyses.

The online version of this article includes the following figure supplement(s) for figure 1:

**Figure supplement 1.** Stimulus and receptive field positions for the V1 "adjacent stimuli" dataset.

**Figure supplement 2.** Eye positions did not differ on single vs. combined stimulus trials (adjacent dataset).

**Figure supplement 3.** Relationship between receptive field (RF) location, stimulus location, spike count model classification , and whether firing rates also correlated with scatter in eye position.

---

focused our analyses on trials in which the monkeys attended to the third stimulus and judged its orientation, that is, attention was consistently directed away from either of the two adjacent Gabor patches that elicited responses in the neurons under study (*Figure 1d*). Trials in which the monkey was required to attend to one or the other of the adjacent Gabor patches were excluded from the analyses, as were incorrectly performed trials.

Any potential contribution of eye movements and/or fixation variation to visually evoked activity was minimized as follows: (1) fixation windows were small, ±0.5° horizontally and vertically, and trials with broken fixations were excluded from further analysis. (2) Any trials with microsaccades during the stimulus presentations (defined as eye velocity exceeding 6 standard deviations above the mean velocity observed during steady fixation; *Engbert and Kliegl, 2003*) were excluded from further analysis. (3) Only a 200 ms period after stimulus onset was analyzed. Our reasoning is that any stimulus-evoked modulation in eye position would have a latency of 150–350 ms (*Engbert and Kliegl, 2003*). This would have limited the consequences of any potential stimulus-evoked fixational modulation to at most only roughly the last 50 ms of the 200 ms spike counting window. In addition, we verified that there actually was no difference in eye position variation based on the stimulus conditions (*Figure 1—figure supplement 2*), so even this slim possibility was not borne out. Finally, we assessed the responses of individual units to ascertain what proportion of units showed a correlation between firing rate and fixational scatter; this proportion was small overall (4–9%) and did not co-vary with the outcomes of the main analyses of the study (see *Figure 1—figure supplement 3* for details).

## Two objects evoke fluctuating activity patterns in V1

We first evaluated the response patterns for evidence of fluctuating activity profiles consistent with multiplexing of information on multistimulus trials. *Figure 2a* illustrates three V1 example units from the adjacent-stimuli dataset, each of which showed spike count distributions on dual stimulus presentations (black lines, 200 ms spike-counting window) that reflected a mixture of the distributions evident on the corresponding single-stimulus presentations (red and blue lines). The dual-stimulus distributions of spike counts are over-dispersed compared to what would be expected if the spikes on dual-stimulus presentations were generated from a similar Poisson process as the single- stimulus presentations, and a tendency for bimodality with modes near the modes for each of the individual stimulus presentations is evident.

While it is visually evident that the spiking responses of these three V1 example units on combined AB stimulus presentations appear drawn from a mixture of the A-like and B-like response distributions, evaluating this systematically across the population requires a formal statistical assessment. We developed such an assessment in our previous work concerning fluctuating activity in the context of encoding of multiple simultaneous stimuli (*Caruso et al., 2018*; *Mohl et al., 2020*; *Glynn et al., 2021*). In particular, we can model the firing rate behavior of neurons when two simultaneous grating stimuli A and B are presented in relation to the firing rates that occur when stimuli A and B are presented individually. We assume that each single-stimulus condition induces Poisson-distributed

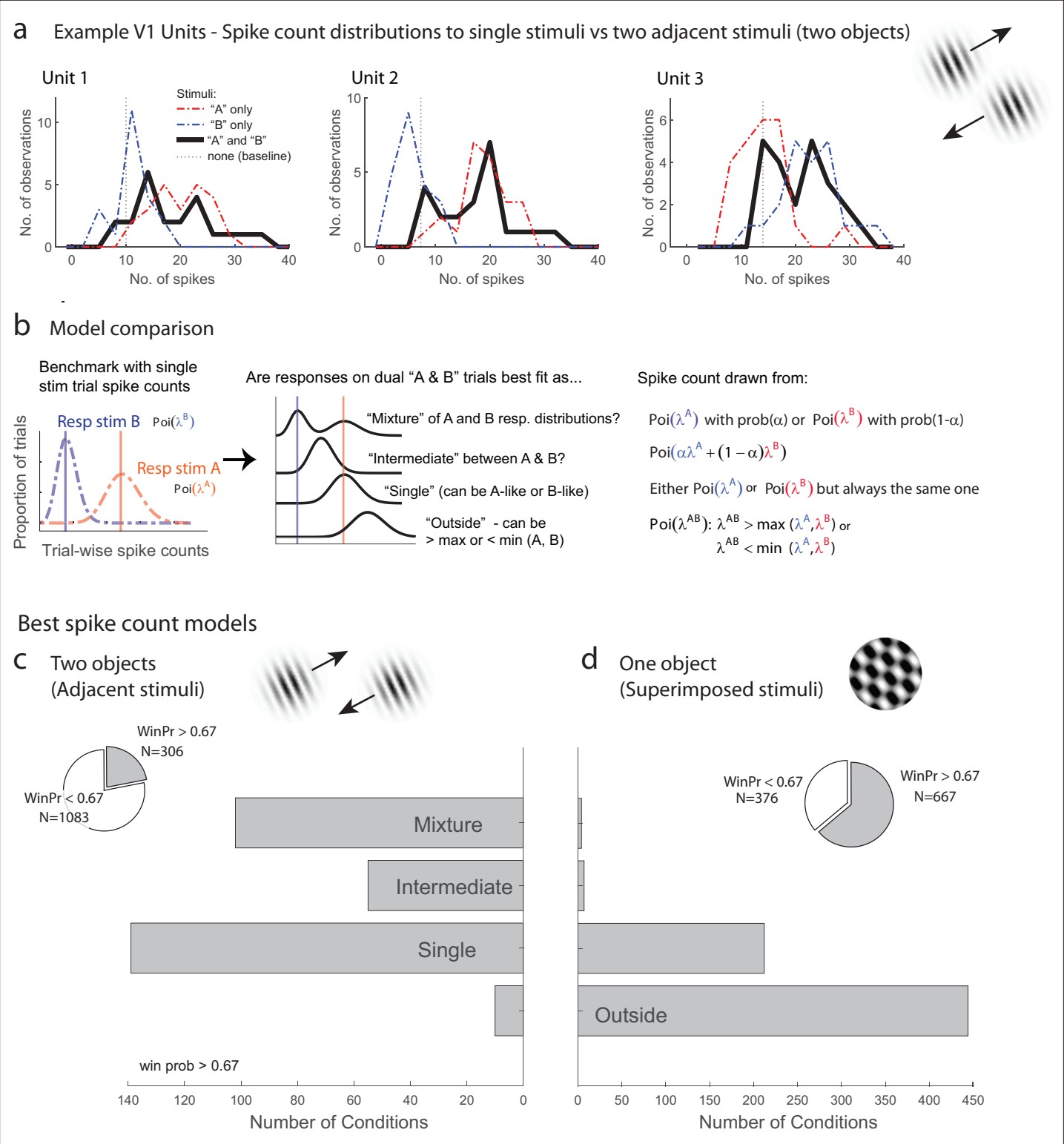

**Figure 2.** Examples of V1 units showing fluctuating activity pattern and formal statistical analysis. (**a**) Distribution of spike counts on single stimuli (red, blue) and dual adjacent stimulus presentations (black) for three units in V1 tested with adjacent stimuli. Spikes were counted in a 200 ms window following stimulus onset. (**b**) Bayesian model comparison regarding spike count distributions. We evaluated the distribution of spike counts on combined stimulus presentations in relation to the distributions observed on when individual stimuli were presented alone. Four possible models were considered as described in the equations and text. Only one case each of the 'single' (B-like) and 'outside' ($\lambda^{AB} > \max(\lambda^{A},\ \lambda^{B})$ is shown. (**c, d**) Best spike count models for the adjacent (**c**) and superimposed (**d**) stimulus datasets, meeting a minimum winning probability of at least 0.67, i.e.,

*Figure 2 continued on next page*

*Figure 2 continued*

the winning model is at least twice as likely as the best alternative. Pie chart insets illustrate proportion of tested conditions that met this confidence threshold. While 'singles' dominated in the adjacent stimulus dataset and 'singles' and 'outsides' dominated in the superimposed stimulus datasets, we focus on the presence of a 'mixtures' as an important minority subpopulation present nearly exclusively in the 'adjacent' stimulus dataset.

The online version of this article includes the following figure supplement(s) for figure 2:

**Figure supplement 1.** Detailed results of the spike count response pattern classification analysis on V1 units for the adjacent (**a**) and superimposed datasets (**b**).

spike counts and we exclude cases where this assumption is violated (see 'Methods' for details). We use a Bayesian model comparison framework to consider four hypotheses concerning the combined AB stimulus presentations (*Figure 2b*): (1) the responses to A and B together appear drawn from the same distribution as either A or B and consistently so on every stimulus presentation, as if the unit responded to only one of the two stimuli ('*single*'). (2) Responses to A and B together appear drawn from a distribution '*outside*' the range spanned by the A and B response distributions; this is the predicted pattern if neurons generally exhibited enhanced responses to combined AB stimuli than either stimulus alone, or if one stimulus strongly suppressed the response to the other. (3) The responses to A and B together are drawn from a single distribution with a mean at an '*intermediate*' value between the A-like and B-like response rates. This is the response pattern that would be expected under theories such as divisive normalization in which the responses of an individual neuron to a more favored stimulus are reduced when other stimuli are also present, but can also represent fluctuating activity on a fast, sub-stimulus-duration timescale, as shown for some neurons in the IT face patch system and inferior colliculus (*Caruso et al., 2018*). (4) The responses to A and B together appear to be drawn from a '*mixture*' of the A-like and B-like response distributions. Mixtures are the category of interest for this analysis as they indicate the presence of activity fluctuations at the stimulus-presentation timescale.

The overall presence of 'mixtures' in V1 differed substantially depending on whether one object or two was presented (the superimposed vs. adjacent grating datasets). *Figure 2c* shows the results for conditions that produced a winning model that was at least twice as likely as its nearest competitor ('win prob >0.67,' the full results are provided in *Figure 2—figure supplement 1*). We found that 'mixtures' were evident in a third of V1 units (33%) when two objects were presented (adjacent gratings, *Figure 2c*), but were very rare when only one 'object' was present (superimposed gratings, *Figure 2d*, 2%). The incidence of 'mixtures' in V1 for the adjacent stimuli was slightly below that observed in the MF face patch in IT cortex (38%) and about half the rate observed in the inferior colliculus (67%; IT and IC data reanalyzed from *Caruso et al., 2018* to use similar winning model criteria as shown here for this study). The remainder of the tested conditions were best explained by the 'single' hypotheses for the adjacent stimuli, indicating winner (or loser)-take-all response patterns, or a blend of 'single' and 'outside' for the superimposed plaid stimuli, indicating the predominance of winner/loser-take-all and either enhancement or suppression in this dataset (see also *Figure 2—figure supplement 1*). This 'single' vs. 'single-or-outside' difference almost certainly stems from differences in the size of the stimuli being presented in these two datasets – typically only one of the two adjacent gratings was located within the classical receptive field of a given V1 unit, whereas this was often not the case for the superimposed dataset. This difference is a side note to our main focus on the fluctuating activity patterns that do occur in V1 in response to multiple objects but not in response to individual objects.

## Possible ways fluctuating activity might be coordinated across the population

Our next question concerns how fluctuating activity patterns are coordinated at the population level, and the implications for preserving information about each of the stimuli that are present. To assess such coordination, we computed Pearson's correlation between the spike count responses observed during presentations a given stimulus combination for pairs of units in each data set (spike count correlation, $r_{sc}$, also commonly called a noise correlation). We begin by discussing the possible results and their interpretation schematically in *Figures 3 and 4*. The overall point is that the activity of pairs

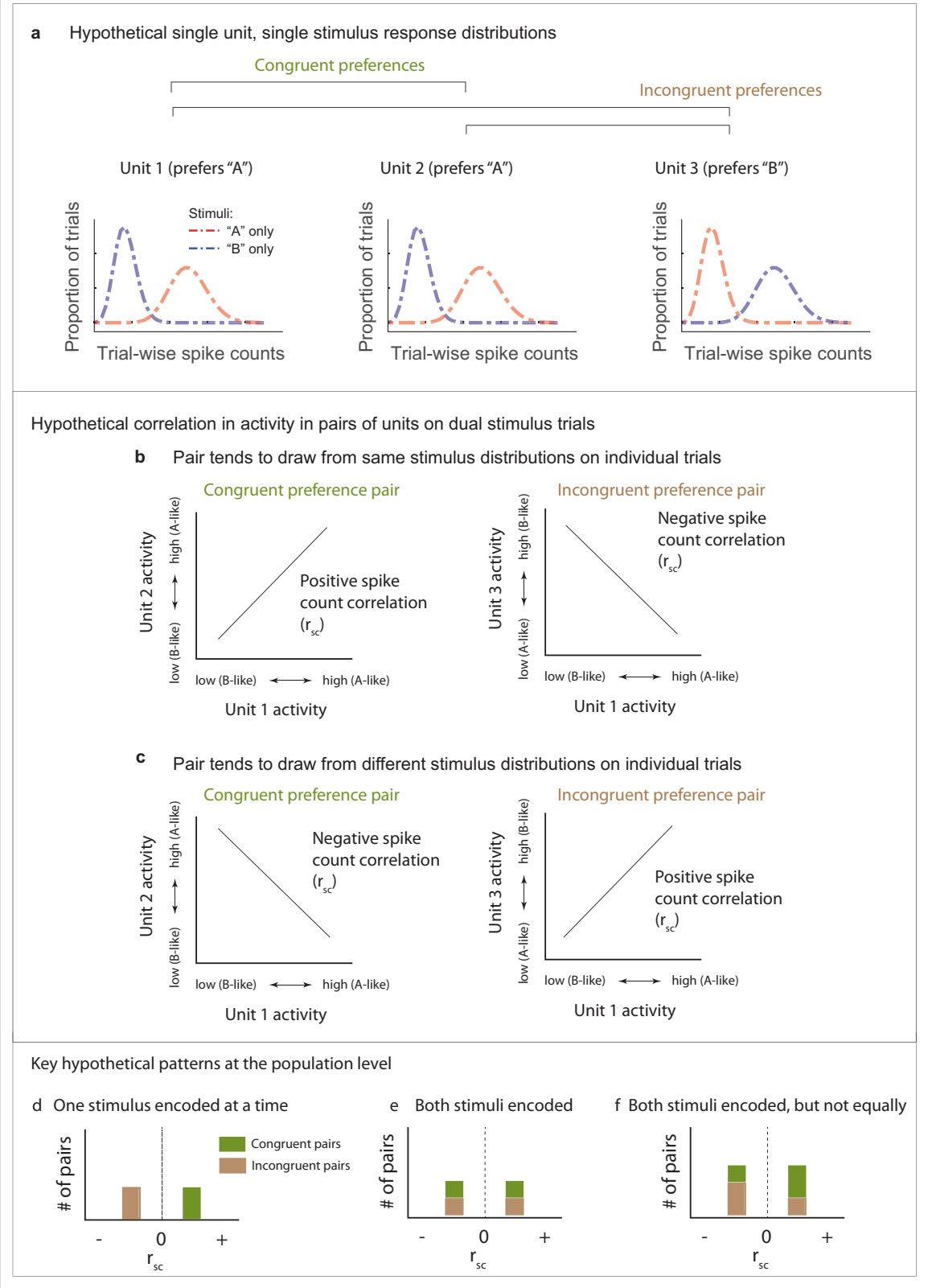

**Figure 3.** Schematic depiction of possible response patterns and resulting correlations. (**a**) Three hypothetical neurons and their possible spike count distributions for single-stimulus presentations. Units 1 and 2 both respond better to stimulus 'A' than to stimulus 'B' ("congruent" preferences). Unit 3 shows the opposite pattern ('incongruent' preferences). (**b, c**) Possible pairwise spike count correlation (Rsc) patterns for these units. Two units that have congruent A vs. B response preferences will show positive correlations with each other if they both show 'A-like' or 'B-like' activity on the same trials

*Figure 3 continued*

(panel **b**, left). In contrast, if one unit prefers 'A' and the other 'B' (incongruent), then A-like or B-like activity in both units on the same trial will produce a negative spike count correlation (panel **b**, right). The opposite pattern applies when units tend to respond to different stimuli on different trials (panel **c**). (**d–f**). Key examples of the inferences to be drawn at the population level from these potential correlation patterns. (**d**) Positive correlations among 'congruent' pairs negative correlations among 'incongruent' pairs would suggest only one stimulus is encoded at the population level at a time. (**e**) If both stimuli are encoded in the population, then both positive and negative correlations might be observed among both congruent and incongruent pairs. (**f**) Both stimuli may be encoded, but not necessarily equally. This example shows a pattern intermediate between the illustrations in (**d**) and (**e**), and is consistent with one of the two stimuli being overrepresented compared to the other. Other possibilities exist as well, including that neurons may could be uncorrelated with one another (not shown), which would also serve to preserve information about both stimuli at the population level.

The online version of this article includes the following figure supplement(s) for figure 3:

**Figure supplement 1.** Details of population-level predictions under different scenarios.

of neurons might be either positively or negatively correlated, and the interpretation of such correlation patterns will depend on the turning preferences of the two neurons in the pair.

*Figure 3* illustrates potential correlation patterns for pairs of several hypothetical neurons, each having a 'mixture' response patterns, but two with a similar or 'congruent' individual stimulus preference (unit 1, unit 2, more spikes elicited by 'A' than 'B' when presented alone) and one with a different stimulus preference compared to the other two (unit 3, more spikes elicited by 'B' than 'A' when presented alone, 'incongruent') (*Figure 3a*). When spike count correlations are computed across trials in which both 'A' and 'B' are presented, four different scenarios (or combinations thereof) could occur. 'Congruent' units 1 and 2 could be positively correlated, suggesting they are encoding the same stimulus on the same trials (i.e., both 'A' or both 'B,' *Figure 3b*, left). Alternatively, they could be negatively correlated, suggesting they are encoding different stimuli on different trials (i.e., one 'A' and the other 'B,' *Figure 3c*, left). Conversely, when considering the spike count correlations between pairs of neurons exhibiting 'incongruent' stimulus preferences (e.g., a 'B' preferring unit 3 vs. the 'A' preferring unit 1), the opposite applies – a positive correlation would be consistent with the two neurons encoding different stimuli in concert (*Figure 3c*, right), and a negative correlation would be consistent with encoding the same stimulus in concert (*Figure 3c*, left). In short, positive vs. negative spike count correlations in response to combined stimuli will have different interpretations depending on whether the two neurons in the pair both respond more vigorously to the same component stimulus or to different component stimuli.

Several key potential patterns of spike count correlations across a population of pairs of neurons are illustrated in *Figure 3d–f*. If the population tends to encode the same stimulus at the same time, then pairs of neurons with congruent preferences will exhibit positive correlations and those with incongruent preferences will exhibit negative correlations (*Figure 3d*). If the population tends to encode both stimuli, then both positive and negative correlations should occur in both pairs with congruent preferences and pairs with incongruent preferences (*Figure 3e*). A third possibility is that both stimuli may be represented at the population level but not evenly so. Such a bias could be reflected by unequal amounts of positive and negative correlations (*Figure 3f*).

It should be noted that it is likely that all spike count correlations between pairs of neurons ride on an overall wave of at least slight positivity due to shared sensitivity to non-stimulus-related factors like overall arousal level or satiety-related signals that might accompany task performance. Thus, the negative- and positive modes of a broad distribution may not be symmetric around zero but slightly shifted toward the positive side.

## With two objects, distinct distributions of positive and negative spike count correlations occur in V1

We now turn to the actual results with these predictions in mind, starting with the example units illustrated in *Figure 2a*. Units 1 and 2 exhibited congruent stimulus preferences: stimulus 'A,' elicited higher spike counts (red line) than stimulus 'B' (blue line) for both. Unit 3 had the opposite (incongruent) preference, with higher spike counts for 'B' than for 'A.' *Figure 4a* shows the activity of each of these units on individual 'A-and-B' stimulus presentations plotted against the others. The pattern of spike count correlation on individual stimulus presentations varied depending on the stimulus tuning preferences, with the pairing between the units with congruent preferences yielding a positive value (0.56, panel d) and the two pairings involving incongruent preferences yielding negative spike

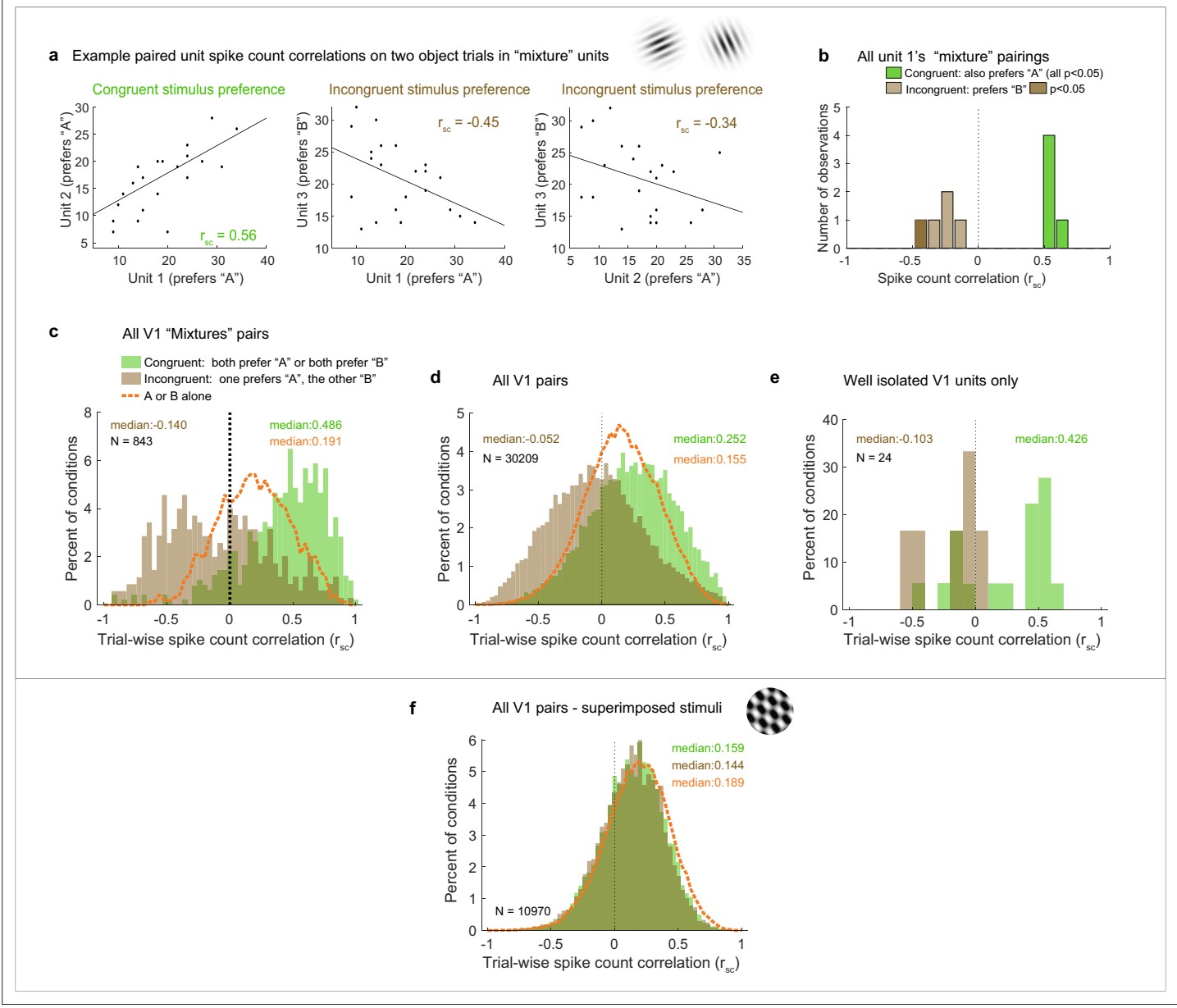

**Figure 4.** Patterns of spike count correlations among pairs of V1 neurons in different subgroups and conditions. (**a**) Example units' correlation patterns (same units as **Figure 2a**). The two units that shared a similar tuning preference ('congruent') exhibited positively correlated spike count variation on individual stimulus presentations for the dual stimulus condition (left), whereas both units 1 and 2 exhibited a negative correlation with the differently tuned ('incongruent') unit 3 (middle and right panels). (**b**) Distribution of $R_{sc}$ values for the other units that were simultaneously recorded with unit 1 and were also classified as 'mixtures,' color coded according to whether the stimulus preference of the other unit was the same as that of unit 1 ('congruent,' green) or different ('incongruent,' brown). All of the 'congruent' pairs exhibited positive correlations, and 5 of 5 were individually significant ($p<0.05$). All of the 'incongruent' pairs exhibited negative correlations, and 1 of 5 was individually significant ($p<0.05$). (**c**) Overall, neural pairs in which both units met the 'mixture' classification showed distinct positive and negative patterns of correlation in response to adjacent stimuli. Positive correlations were more likely to occur among pairs of neurons that responded more strongly to the same individual stimuli ('congruent,' green bars, median $r_{sc} = 0.486$), and negative correlations were more likely to occur among pairs of neurons that responded more strongly to different individual stimuli ('incongruent,' brown bars, median $r_{sc} = –0.14$, $p<0.0001$, see 'Methods'). This bimodal distribution did not occur when only a single stimulus was presented (dashed orange line). (**d, e**) This pattern of results held even when all the unit pairs were considered in aggregate (**d**, 'congruent preference' pairs, median $r_{sc} = 0.252$; 'incongruent preference' pairs, median $r_{sc} = –0.052$, $p<0.0001$), and also occurred for well-isolated single units (**e**). (**f**) However, among pairs recorded during presentation of superimposed gratings, this pattern was not apparent: unit pairs tended to show positive correlations in both cases ('congruent-preference' median $r_{sc} = 0.159$, 'incongruent-preference' median $r_{sc} = 0.144$), and there was little evident difference compared to when a single grating was presented (orange line). See **Figure 4—source data 1** for additional information.

*Figure 4 continued on next page*

*Figure 4 continued*

The online version of this article includes the following source data for figure 4:

**Source data 1.** Median spike count correlations for additional subgroups of the V1 adjacent stimuli dataset.

count correlations (−0.45, −0.34, panels e and f). This pattern is borne out when the full set of pairings involving unit 1 and other units recorded at the same time that also showed 'mixture' response patterns is considered (*Figure 4b*): all the pairings that involved congruent tuning preferences yielded positive correlations, and all of these correlations are individually significant (green bars, p<0.05). In contrast, all the pairings that involve incongruent tuning preferences yielded negative correlations (brown bars); as expected, these are slightly more weakly negative than the congruent pairings are positive, but 1 of 5 reaches individual significance (darker brown, p<0.05).

We next considered the population level (with each pair of units contributing multiple $r_{sc}$ values to the population distribution, one value for each relevant stimulus condition; see 'Methods' for additional details). We first focused on the full set of formally identified 'mixtures' subgroup in the adjacent stimulus dataset (*Figure 4c*), we can see that the pattern observed for the example cells in *Figures 2a and 4a* holds at the population level: neural pairs in which both units responded better to the same individual stimuli tended to have positive correlations with each other ('congruent preferences,' green bars, median 0.486), whereas those that had different ('incongruent') stimulus preferences tended to exhibit negative correlations (brown bars, median –0,14). The spread of values is broad, with many of the pairs of 'incongruent-preference' neurons in particular exhibiting positive values (a point we will return to in Figure 6).

Because of the lack of an adequate population of 'mixture'-classified pairs in the V1 superimposed gratings dataset to compare to the adjacent gratings dataset, we next compared the populations as a whole (*Figure 4d and f*). The patterns are quite different between these two datasets. In the adjacent-stimulus dataset, the overall broad distribution and distinction between congruent-preference and incongruent-preference subgroups holds even when not selecting for 'mixture'

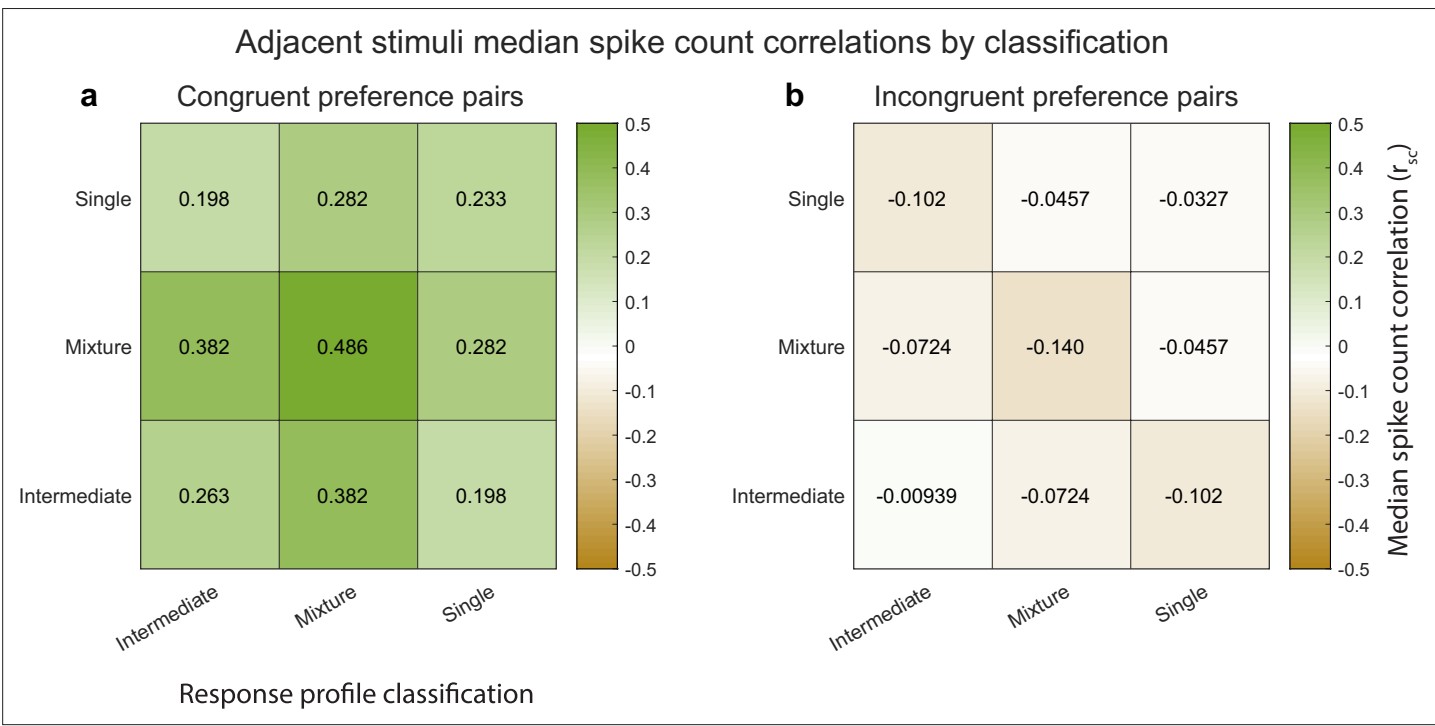

**Figure 5.** Median spike count correlations as a function of congruent-incongruent preference (panel **a** vs. panel **b**) and as a function of the spike count response profile classification resulting from the Bayesian model comparison for the V1 adjacent stimulus dataset. The 'mixture'-'mixture' combinations produced the strongest positive (congruent preference pairs) and strongest negative (incongruent preference pairs) median spike count correlations, but all other combinations also involved positive median correlations for congruent preference pairs and negative median correlations for incongruent preference pairs. See *Figure 4—source data 1* for additional information.

fluctuating patterns (green bars vs. brown bars, median $r_{sc}$ 0.252, –0.052). However, this is much less true of the superimposed-gratings dataset (*Figure 4f*): here, there is very little difference between the congruent-preference and incongruent-preference pairs of neurons (median congruent-preference $r_{sc}$ = 0.159, median incongruent-preference $r_{sc}$ = 0.144), nor is there much difference between the spike count correlations observed on dual-gratings presentations vs. individual grating presentations for this dataset (orange dashed line). In contrast, there is a distinct difference between spike count correlations observed on the dual-stimuli vs. individual-stimulus presentations in the adjacent-stimulus dataset (orange dashed line, *Figure 4c and d*). We verified that this unusual pattern of positive and negative spike count correlations evoked by two objects was not an artifact of multiunit recordings: *Figure 4e* shows the spike count correlation patterns observed for the subset of 24 combinations of well-isolated units recorded simultaneously in the adjacent gratings dataset. While the data is sparse, the overall pattern is consistent with the observations from the full dataset.

We next considered whether this overall pattern was robust to the classification categories emerging from the Bayesian model comparison. While 'mixtures' reflect the strongest evidence for activity fluctuations at a stimulus presentation timescale, activity fluctuations are not fully ruled out among 'singles' and 'intermediates.' For example, if a neuron tended to respond in an 'A-like' fashion on a preponderance of trials but in a 'B-like' fashion on only a few of them, the Bayesian model classifier will rate 'single' as more likely than 'mixture' even though there is some evidence of fluctuation. Relatedly, if a neuron tended to switch between A-like and B-like response patterns more rapidly than the 200 ms stimulus presentation timescale, its overall response pattern would be best described as 'intermediate.' Thus, one might expect the general pattern observed among 'mixtures' to also be present to a lesser degree in these other model categories.

Indeed, this is the case. *Figure 5* illustrates the median spike count correlation by model classification category, for 'congruent-preference' and 'incongruent-preference' pairs of neurons. We excluded the 'outside' category from this analysis as there were too few units that were classified as such. We found that all nine combinations of classifications yielded positive median spike count correlations among 'congruent' preference pairs and negative correlations among 'incongruent' preference pairs. Thus, the overall pattern of results described above does not rest critically on the particular details of the model comparison we implemented here, and is present even among units that could not be formally shown to be fluctuating fully between 'A'-like and 'B'-like response distributions.

Returning to the predictions laid out in *Figure 3d–f*, the implication of congruent-preference units being on average positively correlated and incongruent-preference units being on average negatively correlated from a coding perspective is that V1's representation (among 'mixture' units) may be slightly biased toward one or the other stimulus on each individual stimulus presentation, most closely resembling the schematic depiction in *Figure 3d*. However, the actual data involves a broad distribution with positive spike count correlations also occurring among the incongruent-preference pairs and negative spike count correlations among the congruent-preference pairs. Overall, this is most consistent with the schematic depiction in *Figure 3f*. In short, while the overall pattern of activity among 'mixture' units is biased toward one stimulus over the other on individual stimulus presentations, there are ample cases of units that do not follow this pattern, and these exceptions may be sufficient to preserve information about the other stimulus on any given trial.

To visualize this in another way, we repeated the calculation of Pearson's correlations between pairs of unit conditions classified as mixtures using not the spike counts on each stimulus presentation but an assignment score concerning how 'A'-like vs. 'B'-like the spike count was on an individual stimulus presentation (ranging from 0 to 1; see 'Methods'). Plotted this way, a positive correlation indicates that the two units in the pair tended to exhibit response patterns consistent with the same object at the same time, whereas a negative correlation indicates that the two units tended to exhibit responses consistent with different objects at the same time. The overall pattern in the data is positively skewed (*Figure 6*), but with a long tail on the negative side, consistent with the population of units giving an edge to one stimulus over the other on each individual presentation, but not to the complete exclusion of the other stimulus.

We note that this correlation pattern cannot be accounted for by any obvious confounds. As mentioned previously, all stimulus presentations with microsaccades were excluded from the analyses, limiting the degree to which shared dependence on eye movements could affect the correlation patterns. Furthermore, any variability in fixation position across stimulus presentations might affect the

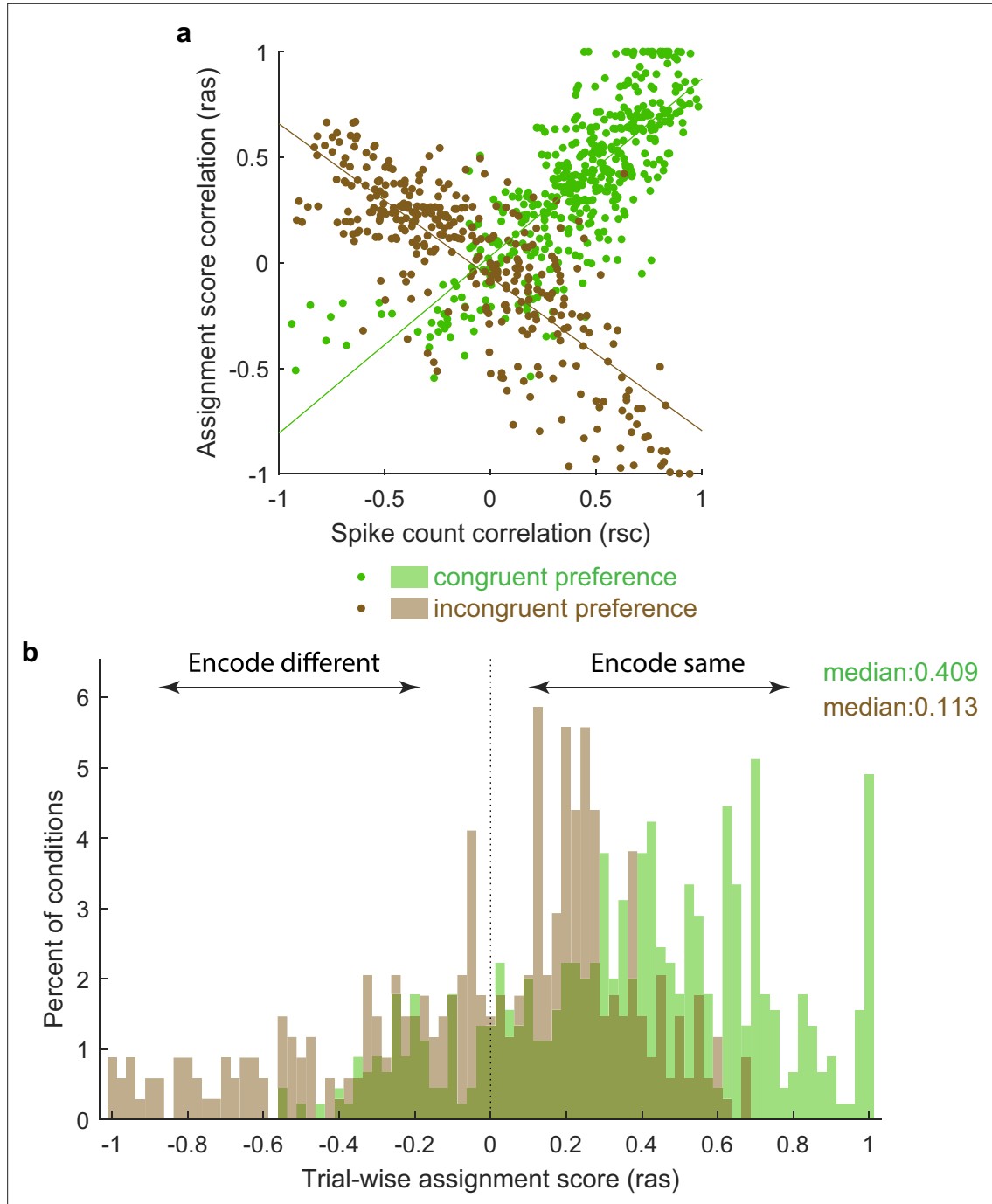

**Figure 6.** Activity fluctuations in 'mixture' pairs of unit conditions are consistent with a bias toward both units in the pair tending to signal the same stimulus at the same time. This analysis involved the Pearson's correlation coefficients computed on assignments scores ($r_{as}$), which take into account whether the response on combined 'AB' stimulus presentations is more 'A-like' vs. 'B-like.' For two units that share a similar preference (e.g., both respond better to A or both respond better to B), this correlation will have the same sign as the spike count correlation (panel **a**, green points, positively sloped best-fit line). For two units that prefer different stimuli, this correlation will be opposite in sign to the spike count correlation (panel **a**, brown points, negatively sloped best-fit line). The overall positive skew in the assignment score correlations for both the 'same' and 'different' preferring unit condition pairs (panel **b**) therefore indicates a bias for the same stimulus at the same time. The negative tail indicates the other stimulus is nevertheless also represented in a (smaller) subpopulation of neurons.

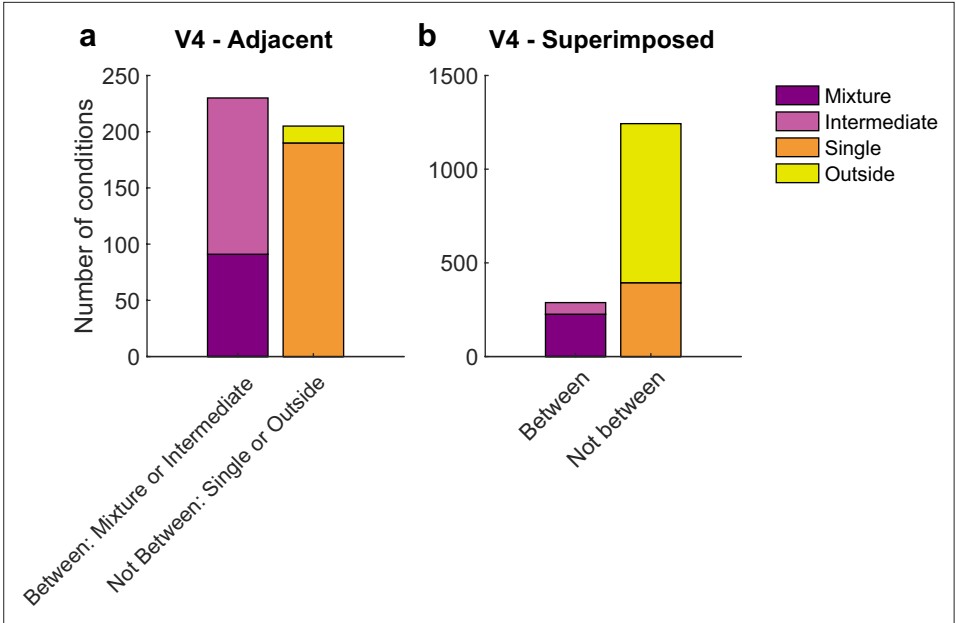

**Figure 7.** Results of the spike count response pattern classification analysis for V4 units. Shown here are classifications for all units regardless of confidence level, and results from gabors and natural images are combined. See *Figure 7—figure supplement 1* for a breakdown by confidence level and for gabors and natural images separately. 'Mixtures' were seen in both datasets, but 'intermediates' were seen primarily in the adjacent-stimulus dataset. These two categories can in principle both contain fluctuating activity, and are grouped here as 'between' (i.e., the average response for dual stimuli for these two categories is between the average responses to single stimuli). As with V1, the relative proportions of 'singles' vs. 'outsides' also differed across these datasets. The combined incidence of these 'not between' categories was higher for the superimposed dataset than for the adjacent dataset.

The online version of this article includes the following figure supplement(s) for figure 7:

**Figure supplement 1.** Detailed results of the spike count response pattern classification analysis on V4 units for the adjacent (**a–c, e**) and superimposed datasets (**d, f**).

assessment of spike count correlations within a particular pair of neurons, but would not be expected to produce (1) a bimodal distribution of spike count correlations at the population level, that (2) occurs especially strongly when two distinct objects are presented. For example, if variability in fixation caused positive correlations between pairs of neurons whose receptive fields were aligned (likely at most a very small subset of our data), this effect should be equally present on both single-stimulus presentations when the stimulus is in those receptive fields and on double-stimulus presentations. Yet, as can be seen in *Figure 5c and d*, the positive extent of the correlations on double-stimulus presentations among 'congruent preference' pairs is higher than is observed on single-stimulus presentations (green bars extend to higher values than the orange curve), and vice versa for the 'incongruent preference' pairs.

## With two objects, distinct distributions of spike count correlations occur in V4

We next assessed V4, which showed both similarities and differences in comparison to V1. Like V1, activity patterns differed considerably in the superimposed vs. adjacent stimuli cases. However, the details of these differences differed: while 'mixtures' were present in both the superimposed and adjacent stimulus conditions in V4, 'intermediates' were more prevalent in the adjacent stimulus case than in the superimposed stimulus case. Given that 'intermediates' could also reflect fluctuations (like 'mixtures' but on a faster-than-stimulus-presentation timescale), we considered both mixtures and intermediates as subcategories of particular interest for the V4 dataset (*Figure 7*).

The patterns of spike count correlations across mixture–mixture pairs in V4 varied considerably based on whether the stimuli were adjacent vs. superimposed and, for adjacent stimuli, whether the

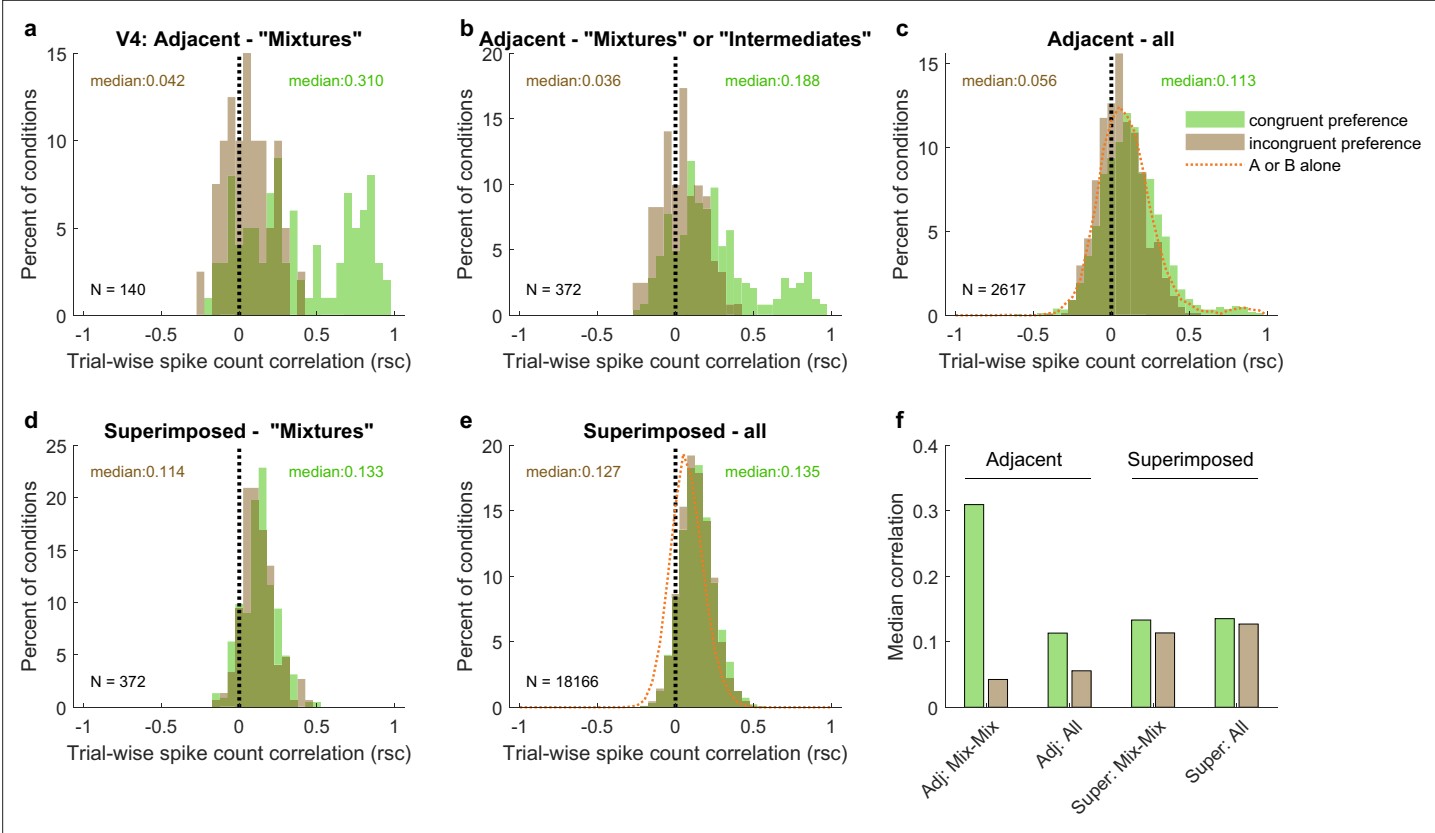

**Figure 8.** Like V1, pairs of V4 units show different patterns of spike count correlations when there are two adjacent stimuli vs. when there is one superimposed stimulus, depending on the tuning preferences of the pair. (**a**) Mixture–mixture pairs for adjacent stimuli (gabors or images), color coded by whether the two units in the pair shared the same or had different tuning preferences. The 'congruent preference' and 'incongruent preference' median correlations differed (p<0.002, see 'Methods'). (**b**) Similar but including intermediate–intermediate pairs since they too may be fluctuating (median difference p<0.0001). (**c**) All unit pairs tested with adjacent stimuli, regardless of classification in the modeling analysis (median difference p<0.0001). Orange line shows the results for single-stimulus presentations. (**d**) Similar to (**a**) but for superimposed stimuli (median difference not significant). (**e**) Similar to (**c**) but for superimposed stimuli (median difference not significant). (**f**) Comparison of median spike count correlations in the adjacent vs. superimposed datasets, color coded by tuning preference.

two units in the pair exhibited the congruent or incongruent stimulus preferences (***Figure 8***). For adjacent mixture–mixture pairs (panel a), the congruent-preferring units again tended to show positive spike count correlations, whereas for incongruent-preferring pairs, the distribution appeared centered around zero. It is unclear whether these pairs are truly uncorrelated or if they might appear uncorrelated due to a negative, stimulus-related, correlation being canceled out by a comparable, globally shared positive correlation that could stem from other factors (e.g., shared reward sensitivity). When intermediate–intermediate patterns are included, the overall pattern of a difference between the congruent-preferring and incongruent-preferring distributions is preserved (panel b), although now the incongruent-preference pairs are slightly positive. This pattern was still present when no selection for response pattern was applied (panel c), and is perhaps best appreciated by comparing the medians of the distributions (***Figure 8f***): there is a distinct difference between the median spike count correlation for same-preference and different-preference pairs for the adjacent dataset. Similar differences in the correlation patterns of congruent-preference vs. incongruent-preference pairs have also been identified in a previous study involving responses of V4 neurons to adjacent gratings (***Verhoef and Maunsell, 2017***).

However, again like V1, when the two stimuli were presented in a superimposed fashion, this difference was no longer evident. This was the case across the whole dataset (***Figure 8e and f***) as well as for mixture–mixture pairs (***Figure 8d and f***), suggesting that when fluctuations do occur for superimposed/bound stimuli, they likely reflect a somewhat different underlying mechanism or purpose than when distinct stimuli are presented.

The preceding spike count correlation analyses capture correlations as a single correlation value per pair of units. This approach is necessary for population-level statistical comparisons and comparison with similar published values in the literature. Reassuringly, the general findings from this approach can also be observed when considering the pattern of responses in a larger set of simultaneously recorded units across individual trials within individual recording sessions. *Figure 9a* shows the results from an individual recording session involving V1 units responding to pairs of adjacent stimuli. Ten units that exhibited 'mixture' response patterns to a particular set of stimuli are shown, with their activity illustrated in color across the 18 trials involving those stimuli. Key observations from this figure match the observations presented previously: (1) individual units show both 'A-like' and 'B-like' (red and blue) response patterns across trials – as expected since we selected 'mixture' units to include in the plot; (2) pairs of units can show correlations with each other (e.g., units 1 and 2 show strongly positively correlated fluctuation patterns, and units 3–5 show positive correlations that are present but somewhat weaker). However, this figure also makes clear that simultaneously recorded units do not correlate perfectly – there is considerable independence in what individual units are doing (compare, for example, units 3–5 with units 1 and 2). The net result of this is that on individual trials some units across the population are responding in an A-like fashion and others are responding in a B-like fashion. *Figure 9b and c* quantify this in a different way – at the cell level and at the trial level, individual cells exhibit some A-like and some B-like responses (as baked in by the selection criteria *Figure 9c*), and on individual trials, some cells exhibit A-like and others B-like responses (*Figure 9b*). This supports our overall interpretation that, at the population level, information about both stimuli is preserved on individual trials.

*Figure 9d and e* illustrate how very different these observed patterns are from two a priori alternative possibilities that would involve loss of information about the two stimuli at the population level. *Figure 9d* captures what one might expect if fluctuations were due to covert shifts of attention – in this case, there might have been strongly correlated fluctuations in activity across all the neurons in the population, not merely individual pairs or small groups. This would appear as vertical stripes of shared blue or shared red across the neural population, indicating that only one stimulus was being encoded at a time. *Figure 9e* captures what one might expect if neurons were not fluctuating at all, but responding to combinations of stimuli by exhibiting normalized or averaged responses intermediate between the responses evoked by either stimulus along – a relatively uniform purple pattern across the neural population. In short, the pattern of responses we observed is quite different from these two alternative 'lossy' possibilities.

## Discussion

The central observations in this article are twofold. First, we identified fluctuating activity patterns in V1, evoked only by combinations of stimuli that are parsed as separate objects. These fluctuations were formally identified using a statistical analysis method benchmarked to the response patterns evoked by each of the stimuli independently (*Caruso et al., 2018*; *Mohl et al., 2020*; *Glynn et al., 2021*). This finding suggests not only that multiplexing of information may be a general characteristic of sensory signals in the brain, but also implicates it in the process of separating vs. grouping of stimuli into objects.

These findings laid the groundwork for our second major question, how fluctuating activity patterns are coordinated across neurons and the implications for coding of stimuli at the population level. We found patterns of spike count correlations that differed substantially from those observed previously, but only when two objects were presented. Single objects (whether individual gratings or two superimposed gratings) yielded correlation patterns very similar to previous reports in the literature (*Cohen and Kohn, 2011*; *Ruff et al., 2016*; *Ruff and Cohen, 2016*), and the correlations did not greatly depend on whether the two units in the pair preferred the same individual stimulus or different ones. In contrast, when two stimuli were presented adjacent to one another other, two distinct distributions emerged based on whether the two units in the pair preferred the same (congruent) individual stimulus (associated with generally positive spike count correlations) vs. different (incongruent) individual stimuli (associated with generally negative spike count correlations in V1 or simply less positive spike count correlations in V4). This pattern was observed in the population as a whole, but was especially pronounced in the subset of units that exhibited 'mixture'-type response patterns indicating

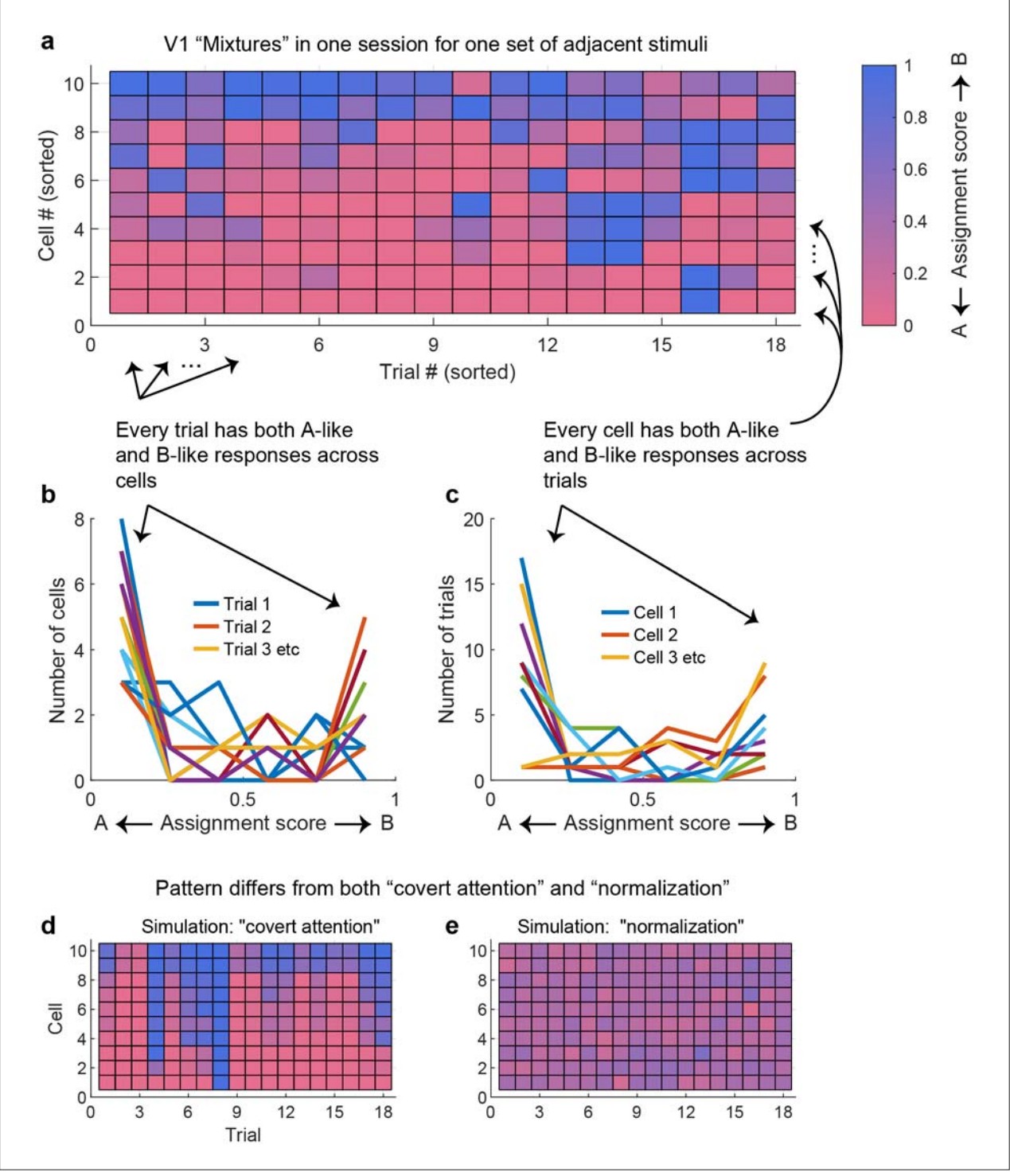

**Figure 9.** At the population level, each stimulus appears to be encoded by at least some units on every trial. (**a**) The activity of 10 simultaneously recorded V1 units on 18 trials in which a particular combination of two adjacent gratings were presented. The activity of each unit was color coded according to how 'A-like' (red) or 'B-like' (blue) the responses were on that trial. Only units for which 'mixture' was the best descriptor of their response patterns are shown (winning probability >0.5, indicating 'mixture' was at least as likely as all other possibilities combined). There are both red and blue squares in every row, supporting the interpretation that these cells exhibited fluctuations across trials. There are also red and blue squares in every column, indicating that on every trial some cells were responding in an 'A-like' fashion and others in a 'B-like' fashion. (**b**) Histogram of the number of cells responding in 'A-like,' 'B-like,' or intermediate levels on each trial (each trace is a separate trial). (**c**) Similar histogram, but indicating the number of trials in which each cell responded in an 'A-like,' 'B-like,' or intermediate firing pattern. (**d**) A simulation of the expected pattern if the observed

*Figure 9 continued on next page*

*Figure 9 continued*

fluctuations chiefly involved covert fluctuations of attention – cells would be expected to show strong correlations with each other and respond in 'A-like' or 'B-like' fashion on the same trials. This simulation was constructed by retaining the cell identity and sets of responses observed for each cell, then instituting a strong correlation between them and shuffling the trials in random order. (**e**) A simulation of the expected pattern if cells were not fluctuating but instead averaging their inputs. This simulation was constructed by assuming that each trial's response represented a draw from a normal distribution with the same mean as the observed distribution (0.34) and a standard deviation of 0.10.

fluctuating across stimulus presentations between the response distributions associated with each of the individual stimuli.

We interpret these observations under the conceptual framework of the challenge that visual cortex faces when representing a visual scene that contains either individual stimuli, combinations of stimuli that bind to form one object, or combinations of stimuli that remain perceptually distinct from each other. The pattern of positive and negative (or less positive) correlations exhibited between pairs of such units is consistent with a population code biased toward one of the two stimuli on any given stimulus presentation, but that preserves information about the other stimulus as well.

It is interesting to note that we observed evidence of multiplexing each stimulus even in V1 where receptive fields are small and the stimuli we used did not themselves typically span more than one receptive field. Put another way, for most of these V1 'mixtures,' the observed fluctuations involved responding vs. not responding rather than fluctuating between two different levels of responding. Thus, the coarseness of tuning did not necessarily pose a problem for the encoding of these particular stimuli in this particular brain area, and yet fluctuations were observed. Thus, the precision of V1's spatial code may not be the limiting factor. Multiplexing is likely to have some as yet unknown characteristic spatial scale that may be determined by the coarsest tuning evident at any stage in the sensory pathway. Future work in which stimuli are systematically varied to manipulate the amount of overlap in the activity patterns evoked in different brain areas by each stimulus alone is needed to answer this question.

The constellation of our findings cannot be easily explained by any obvious alternative explanations. For example, could our focus on the activity patterns of multiunit clusters have impacted the results? If anything, this would be expected to work against the sensitivity of the analyses, if such clusters consisted of individual neurons who were behaving differently from one another. In fact, our findings were broadly similar in the subset of the data that involved well-isolated single units as compared to the full dataset involving multiunit activity. Furthermore, our previous study identifying coding fluctuations in the inferior colliculus and the MF face patch of inferotemporal cortex was conducted on well-isolated single units (*Caruso et al., 2018*). Thus, it seems unlikely that single vs. multiunit isolation significantly impacted our findings.

Could either microsaccades or small differences in the fixation position across trials have impacted the results? As noted earlier, we excluded trials with microsaccades, so such small eye movements are unlikely to have affected the findings, and fixational scatter did not vary by stimulus conditions (*Figure 1—figure supplement 2*). Thus, it is unlikely that variation in fixation position contributed to the difference we observed between two-object vs. fused-object response patterns or the differences between congruent-preference and incongruent-preference pairs of units. Furthermore, as noted above, we observed similar coding fluctuations in two brain areas (the IC and MF face patch) for auditory and large visual stimuli – that is, stimulus conditions that are thought to involve less sensitivity to differences in fixation position than V1 and V4 (*Bremmer, 2000*; *Groh et al., 2001*; *Porter et al., 2006*; *Porter et al., 2007*; *Lehky et al., 2008*; *Maier and Groh, 2010*; *Bulkin and Groh, 2012a*; *Bulkin and Groh, 2012b*; *Merriam et al., 2013*; *Caruso et al., 2018*).

Finally, there is precedent in the literature for differences in the spike count correlation patterns of congruent vs. incongruent-preference pairs: a previous study in V4 (*Verhoef and Maunsell, 2017*) also reported such correlation differences. The general effect size of our V4 results seems to be similar to theirs, particularly when considering the most comparable conditions. The Verhoef and Maunsell study was not designed to identify coding fluctuations, so the most comparable point of comparison in our study would be the pooled results across all model categories (*Figure 8a*). Our analysis focused on units with well-separated responses to the individual stimuli, which is most comparable to the right side of their Fig. 2C; our unit pairs were classified categorically as congruent- or incongruent-preferring rather than on a sliding scale, so it is not immediately apparent how to relate the two studies in that

dimension (y axis in their Fig. 2C). Nevertheless, the approximate difference between the median or mean congruent-preferring vs. incongruent-preferring correlations is reassuringly similar at about 0.06 in our study and a maximum of about 0.12 in theirs. This suggests that the same-preference vs. different-preference correlation patterns observed in the two studies are likely to generalize across different experimental designs.

There has been a rich literature concerning the implications of correlated activity between visually responsive neurons in recent decades. One school of thought considers correlations in the context of the variability of neural firing. Under this 'noise correlation' view, positive correlations have historically been seen as detrimental for encoding information at the population level (*Shadlen and Newsome, 1994*; *Zohary et al., 1994*). Such views have also seen notable refinement and qualification since these early studies (*Romo et al., 2003*; *Averbeck and Lee, 2004*; *Averbeck et al., 2006*; *Moreno-Bote et al., 2014*; *Kanitscheider et al., 2015*; *Kohn et al., 2016*; *Nogueira et al., 2020*; *Kafashan et al., 2021*), including recent work noting that neural variability could be a signal reflecting stimulus uncertainty (*Hénaff et al., 2020*). Arguably closer to the current work is a different school of thought, the temporal correlation hypothesis (*Milner, 1974*; *Gray and Singer, 1989*; *Von Der Malsburg, 1994*; *Singer and Gray, 1995*; *Gray, 1999*). This theory focused on the need to connect the brain's representation of different attributes of a given object together, and proposed that such binding might be mediated through precise synchrony of spikes among neurons responding to the same object. This view, then, sees correlated activity as both useful and specifically relevant to object vision. Studies exploring this hypothesis have, however, primarily focused on within-trial temporal synchrony of spikes on the order of milliseconds, whereas the noise correlation literature has focused on spike counts in the domain of hundreds of milliseconds and analysis at the level of the ensemble of trials or stimulus presentations. By evaluating spike count variation at the level of stimulus presentations and comparing the results as a function of the number of stimuli/objects, the present work forges a bridge between these two areas of the literature.

Our findings also suggest reconsideration of two other key processes in visual neuroscience: selective attention and normalization. Selective attention refers to the fact that perceptual awareness is not equal across all stimuli present in a sensory scene. Selective attention can be controlled through 'top-down' means, such as via tasks in which participants are cued to focus on one stimulus and ignore others. Indeed, the monkeys were performing just such a task in our adjacent V1 dataset, and thus were in theory ignoring the stimuli whose responses we studied here. But even with correct task performance, top-down control of attention is imperfect. Might the fluctuating responses we observed be due to covert shifts of attention from one of the supposedly unattended stimuli to the other (which could contribute to the observed activity patterns; *Ecker et al., 2016*; *Engel et al., 2016*; *Denfield et al., 2018*)? We think not. If this were the case, then the neurons should have fluctuated in more perfect harmony with one another. As shown quantitatively in *Figure 6* and qualitatively in *Figure 9*, although there is a bias in which stimulus is 'capturing' the response patterns on individual trials, there remains a substantial portion of the neural population that is responding to the other stimulus. And the observed pattern of fluctuating activity is very different from a simulation of covert attention (*Figure 9a* vs. *Figure 9d*). While further work on this question is needed, we think it is worth noting that the patterns of activity that we observed can in principle support preservation of information about all stimuli in the scene. Processes involved in selective attention might contribute to the creation of biases within this representation or could act at later stages on the information preserved within these representations to enhance awareness of one or more of the represented stimuli.

Previous findings from the existing literature on attention and related areas are consistent with this new view, and could easily be evaluated anew using the approach we described here. Trial-averaged neural responses to attended and unattended stimuli can often be modeled as a weighted combination of the responses to those stimuli when presented alone (*Boynton, 2009*; *Lee and Maunsell, 2009*; *Reynolds and Heeger, 2009*; *Ni et al., 2012*; *Ni and Maunsell, 2017*; *Verhoef and Maunsell, 2017*; *Ni and Maunsell, 2019*; *Lee and Maunsell, 2009*; *Ni et al., 2012*; *Ni and Maunsell, 2017*; *Verhoef and Maunsell, 2017*; *Ni and Maunsell, 2019*). Such averaging responses are seen not only in attention paradigms but in other contexts as well (e.g., see also *Xiao et al., 2014*; *Xiao and Huang, 2015*) and are generally referred to as normalization. Importantly, these reports have generally concerned responses pooled across trials. Trial-wise spike count distribution models such as those used here and/or faster subtrial analyses such as those we have introduced in previous work (*Caruso*

*et al., 2018*; *Glynn et al., 2021*) might indicate that such apparently averaging responses actually indicate fluctuations occurring on either the stimulus presentation or sub-stimulus presentation timescales, and not a true stable average (e.g., *Figure 9a* vs. *Figure 9e*).

That 'normalization' may not involve a fixed, stable operation that is constant across trials has recently garnered considerable interest. For example, several important recent studies have begun to explore how recurrent circuit mechanisms might implement dynamic fluctuations in neural activity (*Heeger and Mackey, 2019*; *Heeger and Zemlianova, 2020*) and have postulated that shifts in the balanced excitation and inhibition that is thought to underlie normalized average responses when two stimuli are presented might contribute to sizeable positive or negative spike count correlations (*Verhoef and Maunsell, 2017*). Finally, recent work by Coen-Cagli and colleagues proposes a method of assessing normalization strength on individual trials and demonstrated a connection between the neural responses that are well-described under a normalization model and the level of variability of firing that they show (*Coen-Cagli and Solomon, 2019*; *Weiss et al., 2022*).

Returning to the topic of attention, recent work suggests a perceptual tie to the findings we report here. The likelihood of detecting a brief near-threshold visual stimulus varies with the phase of the brain wave oscillations at the time the stimulus is presented (*Busch et al., 2009*; *Busch and VanRullen, 2010*; *Vanrullen et al., 2011*; *Fiebelkorn et al., 2013*; *Fiebelkorn et al., 2018*; *Helfrich et al., 2018*; *Fiebelkorn and Kastner, 2019*; see also *Engel et al., 2016*). This might reflect a perceptual consequence of a brain mechanism in which neurons are slightly biased toward representing some stimuli in the visual scene over others in a naturally occurring oscillatory fashion. Such bias was evident in the responses observed here, although we did not deploy a task to assess any potential connection to behavior. In our previous study (*Caruso et al., 2018*), we found that the LFP signal prior to stimulus onset was predictive of whether neurons would 'pick' A vs. B on a given trial. Future work will be needed to ascertain whether a similar phenomenon occurs in V1 or V4.

Finally, it is worth noting here that considering how the brain preserves information about *two* visual stimuli presented is still a far cry from understanding how the myriad details present in a natural scene are encoded. When the number of objects gets too great, it is unlikely that neurons can fluctuate between all of them, and this is likely to have consequences for perception, perhaps accounting for well-known limits on the number of objects we can perceive, attend to, and remember (e.g., *Miller, 1956*; *Whitney and Levi, 2011*; *Henry and Kohn, 2020*). Future studies incorporating many stimuli and investigating how this changes the pattern of fluctuating activity and correlations between units are needed to shed light on how our brains operate outside the rarefied environment of the laboratory.

## Methods

### Electrophysiological recordings and visual stimuli

The full experimental procedures are described in *Ruff et al., 2016* and *Ruff and Cohen, 2016* and summarized below. All animal procedures were approved by the Institutional Animal Care and Use Committees of the University of Pittsburgh and Carnegie Mellon University (Protocol #: 20067560 PHS Assurance Number: D16-00118). Each of the datasets consisted of multielectrode recordings

**Table 2.** Trial counts for included sessions.

The values reported are calculated for individual recording sessions for which at least one triplet was included for the analysis; the numbers of trials are the same for all simultaneously recorded units within a session. The values for 'A' and 'B' trials indicate the values for either A or B; that is, there were on average 21 'A' trials and 21 'B' trials for each triplet in the adjacent V1 dataset.

| Stimuli | Brain area | Number of 'A' and 'B' trials | | | | Number of 'AB' trials | | | |
|---|---|---|---|---|---|---|---|---|---|
| | | Mean | SD | Min | Max | Mean | SD | Min | Max |
| Adjacent | V1 | 21.0 | 12.3 | 6 | 56 | 17.8 | 12.8 | 6 | 59 |
| | V4 | 72.8 | 30.3 | 5 | 136 | 72.2 | 30.7 | 6 | 132 |
| Superimposed | V1 | 25.4 | 15.4 | 7 | 74 | 23.3 | 12.1 | 7 | 64 |
| | V4 | 131.3 | 42.0 | 20 | 196 | 184.5 | 64.6 | 20 | 270 |

from two adult male rhesus monkeys for each brain area (*Tables 1 and 2*). Recordings were made using chronically implanted a 10 × 10 microelectrode arrays (Blackrock Microsystems) in V1 and 6 × 8 arrays in V4 (*Figure 1a*). The electrode shafts were 1 mm long, and the minimum distance between the nearest electrodes was 400 μm. In some sessions, recordings were also made using other electrodes in areas MT and 7a, but these data are not included in the current analyses.

The visual stimuli and behavioral experiment for the superimposed stimulus dataset are fully described in *Ruff et al., 2016*. Monkeys were rewarded for passively viewing individual or superimposed orthogonal drifting gratings, positioned to span the receptive fields of the entire population of neurons under study (size range: 2.5–7°). As noted above, the fixation windows were ±0.5° horizontally and vertically, and stimulus presentations with microsaccades (defined as eye velocity exceeding 6 standard deviations above the mean velocity observed during steady fixation; *Engbert and Kliegl, 2003*) were excluded from further analysis. In the full dataset, multiple contrast levels were presented, most of which were not included for analysis in this study. Here, we included trials in which one grating had a contrast of 0 (i.e., was not visible) and the other had a contrast of 0.5 ('A'-alone and 'B'-alone cases) or both gratings had a contrast of 0.5 ('AB'). In most sessions, each stimulus lasted for 200 ms; a few sessions with 1000 ms stimuli were also included but only the first 200 ms were analyzed, that is, spikes were counted in a 200 ms window after stimulus onset for all the analyses in this study. This spike counting window was offset by the typical response latency for the region under study, that is, 30–230 ms for V1 and 50–250 ms for V4.

The visual stimuli and behavioral experiment for the V1 adjacent stimulus dataset are fully described in *Ruff and Cohen, 2016*. The animals performed a motion direction change detection task in which they were cued in blocks of trials to attend to small drifting Gabor patches (~1°) in various locations and respond when the orientation of the attended location changed. In this study, we analyzed trials in which attention was directed to a Gabor patch located in the hemisphere ipsilateral to the recorded V1 neurons (i.e., well away from those neurons' receptive fields, see below). On these trials, two unattended Gabor patches were presented in close proximity to each other within the area covered by the receptive fields of the recorded V1 neurons – these receptive fields were approximately 3° eccentric and had classical receptive field diameters typically estimated to be <1° of visual angle. These patches were centered 2.5–3.5° eccentrically and each stimulus typically subtended 1° of visual angle (see *Ruff and Cohen, 2016* Fig. 1B for a sketch, reproduced here in *Figure 1—figure supplement 1*). The patches had the same orientation but drifted in opposite directions and were flashed on for 200 ms and off for 200–400 ms. We analyzed responses to all stimuli before the orientation change, excluding the first stimulus in every trial. Again, only correctly performed trials with no microsaccades during stimulus presentations were included for analysis. As noted previously, monkeys were required to maintain fixation within ±0.5°, and typically fixation was more precise than required; see *Figure 1—figure supplement 2* for fixational scatter in an example session and across sessions. Correlations between firing rates and scatter in fixation position were assessed for the dual-stimulus trials using the component of eye position that lay along a line connecting the two stimulus locations chosen for the recording session (see *Figure 1—figure supplement 3* for results).

The adjacent stimulus dataset for V4 involved two types of stimuli, small drifting Gabor patches as above or natural images of animals or common objects, from *Long et al., 2018*. Results for the two types of stimuli were combined for the main analyses presented in this article (*Figures 7 and 8*), and are broken out separately in *Figure 7—figure supplement 1*. The monkeys performed a fixation task.

## Analysis of spike count distributions

The full description of the statistical evaluation of spike count distributions on combined stimulus presentations can be found in *Caruso et al., 2018*; *Mohl et al., 2020*. Briefly, we deployed a Bayesian procedure for modeling the distribution of spike counts in response to combined stimuli. Assuming that the spike counts corresponding to condition A and condition B are both Poisson-distributed with the rate parameters $\lambda^A$ and $\lambda^B$, respectively (and excluding exceptions, see below), the four hypotheses for the spike count distributions for condition AB consist of

1. 'Single': A single Poisson distribution Poi($\lambda^{AB}$), where $\lambda^{AB}$ exactly equals either $\lambda^A$ or $\lambda^B$.
2. 'Outside': A single Poisson distribution Poi($\lambda^{AB}$), where $\lambda^{AB}$ lies outside the range between $\lambda^A$ and $\lambda^B$.

3. 'Intermediate': A single Poisson distribution Poi($\lambda^{AB}$), where $\lambda^{AB}$ lies inside the range between $\lambda^A$ and $\lambda^B$.
4. 'Mixture': A mixture of the two single-stimulus distribution with an unknown mixing rate α: α Poi($\lambda^A$) + (1-α) Poi($\lambda^B$).

For each 'triplet' or combination of A, B, and AB conditions, the hypothesis with the highest posterior probability was selected, based on the intrinsic Bayes factor (*Berger and Pericchi, 1996*) and using the equal prior (¼) among the four hypotheses, Jeffrey's prior (*Berger and Pericchi, 1996*) for the Poisson rate parameters, and a uniform prior in [0, 1] for the mixing weight α.

Only the triplets satisfying two exclusion criteria are used: (1) the single-stimulus distributions follow Poisson distributions, and (2) the single-stimulus rate parameters $\lambda^A$ and $\lambda^B$ are substantially separated. The first criterion was tested using Monte Carlo p-value calculation for a chi-square goodness-of-fit test (p>0.10), and the second criterion was tested by whether the intrinsic Bayes factor of the model $\lambda^A \neq \lambda^B$ is more than three times higher than that of the model $\lambda^A = \lambda^B$. These exclusion criteria were applied to all the analyses in the article, even those that did not build specifically on this model classification, to ensure that comparisons between subpopulations of the data were not affected by differences in data selection criteria.

The numbers of trials involved for the different datasets are provided in *Table 2*. The trial counts were adequate to provide accurate model identification according to our previous simulations. Depending on the separation between $\lambda^A$ and $\lambda^B$, we previously found that model identification accuracy in simulations is high for trial counts as low as 5 'AB' trials, and plateaus near ceiling around 'AB' trial counts of about 10 trials and above – that is, below the mean trial counts available here for all datasets (see Figure 4 of *Mohl et al., 2020*).

## Correlation analysis

We calculated spike count correlations between pairs of units recorded at the same time in the same experiment. The Pearson correlation coefficient was calculated on the spike counts for each presentation of each relevant stimulus combination. Stimulus presentations in which one or both units in the pair exhibited an 'outlier' response, that is, more than 3 standard deviations from the mean, were excluded from the analysis. The spike count correlations for particular unit pairs for different stimulus combinations were included in the population analyses as separate observations and were not averaged together. For example, in the V1 adjacent stimulus dataset, pairs were typically tested with two separate adjacent stimulus combinations, differing in the direction of motion, potentially yielding two values of the spike count correlation (assuming both conditions passed the Poisson and response-separation exclusion criteria noted above). Similarly, V4 neurons tested with different combinations of drifting gabors and/or images contributed values of spike count correlations for each stimulus set to the population.

## Congruent or incongruent preference

Preference of a unit for a particular stimulus was determined by higher spike counts. Unit pairs that both exhibited more spikes in response to stimulus A than to B, or both exhibited more spikes in response to stimulus B than to A, were defined as 'congruent preference.' Unit pairs in which one responded with more spikes to A and the other with more spikes to B were defined as 'incongruent preference'.

## Comparison of distributions of spike count correlations

The medians of the 'congruent preference' vs. 'incongruent preference' distributions of spike count correlations were statistically compared using Monte Carlo methods in which the same/different preference assignments were randomly shuffled and the medians recalculated 10,000 times. When the true difference between the medians was greater than any of the shuffled versions, the p-value can be said to be less than 1/10,000 or 0.0001.

## Acknowledgements

The authors thank Valeria C Caruso, Yunran Chen, Jeffrey T Mohl, Meredith N Schmehl, and Shawn M Willett for helpful comments on the analysis and manuscript. This work was supported by the National Institutes of Health grant nos. R00EY020844 (MRC), R01EY022930 (MRC); Core Grant P30 EY008098s (MRC); R01DC013906 (JMG, STT); and R01DC016363 (JMG, STT); and by support to MRC from the McKnight, Whitehall, Sloan and Simons Foundations.

## Additional information

### Competing interests

Jennifer M Groh: Reviewing editor, eLife. The other authors declare that no competing interests exist.

### Funding

| Funder | Grant reference number | Author |
|---|---|---|
| National Institutes of Health | R00EY020844 | Marlene R Cohen |
| National Institutes of Health | R01EY022930 | Marlene R Cohen |
| National Institutes of Health | Core Grant P30 EY008098s | Marlene R Cohen |
| National Institutes of Health | R01DC013906 | Jennifer M Groh |
| National Institutes of Health | R01DC016363 | Jennifer M Groh |
| McKnight Endowment Fund for Neuroscience | | Marlene R Cohen |
| Whitehall Foundation | | Marlene R Cohen |
| Alfred P. Sloan Foundation | | Marlene R Cohen |
| Simons Foundation | | Marlene R Cohen |
| National Institutes of Health | R01 DC013906 | Surya T Tokdar |
| National Institutes of Health | R01 DC016363 | Surya T Tokdar |

The funders had no role in study design, data collection and interpretation, or the decision to submit the work for publication.

### Author contributions

Na Young Jun, Data curation, Software, Formal analysis, Validation, Visualization, Methodology, Writing – original draft, Writing – review and editing; Douglas A Ruff, Conceptualization, Resources, Data curation, Software, Validation, Investigation, Methodology, Writing – review and editing; Lily E Kramer, Brittany Bowes, Data curation, Investigation; Surya T Tokdar, Conceptualization, Resources, Software, Formal analysis, Supervision, Funding acquisition, Validation, Visualization, Project administration, Writing – review and editing; Marlene R Cohen, Conceptualization, Resources, Data curation, Software, Supervision, Funding acquisition, Investigation, Methodology, Project administration, Writing – review and editing; Jennifer M Groh, Conceptualization, Data curation, Software, Formal analysis, Supervision, Funding acquisition, Validation, Visualization, Methodology, Writing – original draft, Project administration, Writing – review and editing

### Author ORCIDs

Na Young Jun http://orcid.org/0000-0002-8841-3947
Jennifer M Groh http://orcid.org/0000-0002-6435-3935

### Ethics

All animal procedures were approved by the Institutional Animal Care and Use Committees of the University of Pittsburgh and Carnegie Mellon University: Protocol #: 20067560 PHS Assurance Number: D16-00118.

### Decision letter and Author response

Decision letter https://doi.org/10.7554/eLife.76452.sa1
Author response https://doi.org/10.7554/eLife.76452.sa2

## Additional files

### Supplementary files

• Transparent reporting form

• Source data 1. Source data for the figures and analyses in this article are included as a zip file. The file and folder names are informative regarding which analyses they relate to. Some analyses are based on multiple runs of the modeling code, with slight variations due to the probabilistic nature of the analysis.

### Data availability

Source data is provided in a zip file; the individual files have informative names that indicate what they contain and thus to what figures they are linked.

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
