## [Editor Report]

The authors report that neurons in V1 and V4 multiplex information of simultaneously presented objects. A combination of multi-single unit recordings, statistical modelling of neuronal responses and neuronal correlations analyses argues in favor of their claims. Pairs of neurons having similar object preferences tended to be positively correlated when both objects were presented, while pairs of neurons having different object preferences tended to be negatively correlated and these patterns and others suggest that information about the two objects is multiplexed in time. These results are of broad interest to the field, as they shed new light on the "binding" problem and highlight the importance of underexplored features of cortical activity for neural coding.

---

## [Decision Letter]

**Decision letter after peer review:**

Thank you for submitting your article "Coordinated multiplexing of information about separate objects in visual cortex" for consideration by *eLife*. Your article has been reviewed by 3 peer reviewers, one of whom is a member of our Board of Reviewing Editors, and the evaluation has been overseen by Joshua Gold as the Senior Editor. The following individual involved in the review of your submission has agreed to reveal their identity: Ruben Coen-Cagli (Reviewer #3).

Essential revisions:

While this paper shows some intriguing results, we feel that there are a lot of open questions that need to be addressed before convincing evidence of multiplexing can be established. These points are discussed below:

1. The best spike count model shown in Figure 2C is confusing. It seems that the number of "conditions" is a small fraction of the total number of conditions (and neurons?) that were tested. Supplementary Figure 1 provides more details (for example, the "mixture" corresponds to only 14% of total cases), but it is still confusing (for example, what does WinProb>Min mean?). From what I understood, the total number of neurons recorded for the Adjacent case in V1 is 1604, out of which 935 are Poisson-like with substantially separated means. Each one has 2 conditions (for the two directions), leading to 1870 conditions (perhaps a few less in case both conditions were not available). I think the authors should show 5 bar plots – the first one showing the fraction for which none of the models won by 2/3 probability, and then the remaining 4 ones. That way it is clear how many of the total cases show the "multiplexing" effect. I also think that it would be good to only consider neurons/conditions for which at least some minimum number of trials are available (a cutoff of say ~15) since the whole point is about finding a bimodal distribution for which enough trials are needed.

2. More RF details need to be provided. What was the size of the V1 RFs? What was the eccentricity? Typically, the RF diameter in V1 at an eccentricity of ~3 degrees is no more than 1 degree. It is not enough to put 2 Gabors of size 1 degree each to fit inside the RF. How close were the Gabors? We are confused about the statement in the second paragraph of page 9 "typically only one of the two adjacent gratings was located within the RF" – we thought the whole point of multiplexing is that when both stimuli (A and B) are within the RF, the neuron nonetheless fires like A or B? The analysis should only be conducted for neurons for which both stimuli are inside the RF. When studying noise correlations, only pairs that have overlapping RFs such as both A and B and within the RFs of both neurons should be considered. The cortical magnification factor at ~3-degree eccentricity is 2-2.5mm/degree, so we expect the RF center to shift by at least 2 degrees from one end of the array to the other.

3. Eye data analysis: the reviewers are concerned that this could potentially be a big confound. Removing trials that had microsaccades is not enough. Typically, in these tasks the fixation window is 1.5-2 degrees, so that if the monkey fixates on one corner in some trials and another corner in other trials (without making any microsaccades in either), the stimuli may nonetheless fall inside or away from the RFs, leading to differences in responses. This needs to be ruled out. We do not find the argument presented on pages 18 or 23 completely convincing, since the eye positions could be different for a single stimulus versus when both stimuli are presented. It is important to show that the eye positions are similar in "AB" trials for which the responses are "A" like versus "B" like, and these, in turn, are similar to when "A" and "B" are presented alone.

Relatedly, more details on the detection of microsaccades and threshold values for inclusion (relative to stimulus and RF sizes) should be provided, given how central a role they might play. In particular, the authors state in Discussion that small residual eye movements would inflate response variability in all stimulus conditions. This is correct, but because of the stimulus design, it is possible (likely?) that the effects are quite different for segregated stimuli versus superimposed and single-stimulus conditions. Furthermore, the difference might be precisely in the direction of the effects reported. That is because segregated stimuli are spatially separate, and each stimulus only covers some receptive fields in the recording, it is possible that eye movements would bring inside the RF a different stimulus in each trial. In addition to producing bimodal response distributions for individual neurons, this would also induce positive correlations for pairs with the same preference and negative correlations for pairs with opposite preferences. On the other hand, in the superimposed condition where the stimulus is large enough to cover all RFs, at most eye movements would bring (part of) the stimulus inside versus outside the RF across trials, therefore contributing to positive noise correlations for all pairs (i.e., to shifts from stimulus-driven to spontaneous activity, in the extreme case). In our opinion, the main concern raised in the public review deserves more in-depth analysis, of the data and/or simulations, for us to be convinced that eye movements truly do not play a role. Conversely, if they do, it may be worth discussing the possibility that they are part of the mechanism underlying the proposed coding scheme.

4. Figures 5 and 6 show that the difference in noise correlations between the same preference and different preference neurons remains even for non-mixture type neurons. So, although the reason for the particular type of noise correlation was given for multiplexing neurons (Figure 3 and 4), it seems that the same pattern holds even for non-multiplexers. Although the absolute values are somewhat different across categories, one confound that still remains is that the noise correlations are typically dependent on signal correlation, but here the signal correlation is not computed (only responses to 2 stimuli are available). If there is any tuning data available for these recordings, it would be great to look at the noise correlations as a function of signal correlations for these different pairs. Another analysis of interest would be to check whether the difference in the noise correlation for simply "A"/"B" versus "AB" varies according to neuron pair category. Finally, since the authors mention in the Discussion that "correlations did not depend on whether the two units preferred the same stimulus or different", it would be nice to explicitly show that in figure 5C by showing the orange trace ("A" alone or "B" alone) for both same (green) and different (brown) pairs separately.

5.We are confused about the nature of Poisson models. If we are correct, the Poisson(a+b) is the sum of the two Poisson(a) and Poisson(b), that is, Poisson(a+b) = Poisson(a) + Poisson(b). Then, the mixture and intermediate models are very similar, identical if a*λ_A and (1-a)*λ_B happen to be integer numbers.

6. It is unclear why the 'outside' model predicts responses outside the range if neurons were to linearly sum the A and B responses.

7. It is also unclear why the 'single' hypothesis would indicate a winner-take-all response. If we understand correctly, under this model, the response to A+B is either the rate A or B, but not the max between λ_A and λ_B. Also, this model could have given an extra free parameter to modulate its amplitude to the stimulus A+B.

8. The concept of "coarse population coding" can be misleading, as actual population coding can represent stimulus with quite good precision. The authors refer to the broad tuning of single cells, but this does not readily correspond to coarse population coding. This could be clarified.

9. As a complement to the correlation analysis, one could check whether, on a trial-by-trial basis, the neuronal response of a single neuron is closer to the A+B response average, or to either the A or B responses. This would clearly indicate that the response fluctuates between representing A or B, or simultaneously represents A+B. I am trying to understand why this is not one of the main analyses of the paper instead of the correlation analysis, which involves two neurons instead of one.

10. In the discussion about noise correlations, the recent papers Nogueira et al., J Neuroscience, 2020 and Kafashan et al., Nat Comm, 2021 could be cited. Also, noise correlations can also be made time-dependent, so the distinction between the temporal correlation hypotheses and noise correlations might not be fundamental.

11. It would be interesting to study the effect of contrast on the mixed responses. Is it reasonable to predict that with higher contrast the mixture responses would be more dominant than the single ones? This could be the case if the selection mechanism has a harder time suppressing one of the object responses. This would also predict that noise correlations will go down with higher contrast.

12. What is the time bin size used for the analysis? Would the results be the same if one focuses on the early time responses or on the late time responses? At least from the units shown in Figure 2, it looks that there is always an object response that is delayed respect to the other, so it would seem interesting to test noise correlations in those two temporal windows.

[Editors' note: further revisions were suggested prior to acceptance, as described below.]

Thank you for resubmitting your work entitled "Coordinated multiplexing of information about separate objects in visual cortex" for further consideration by *eLife*. Your revised article has been evaluated by Joshua Gold (Senior Editor) and a Reviewing Editor.

The manuscript has been improved and many of the reviewers' concerns have been addressed. However, some major issues remain. Although we typically avoid repeated revise/resubmit cycles, we believe that it is important for these issues to be addressed in a new revision. Specifically, upon discussion the reviewers unanimously remained concerned about the possibility that at least some of your results could be accounted for by subtle differences in eye movements. The new analyses related to that issue were appreciated but considered inadequate.

As detailed below, we would like you to provide:

1. More information about whether the electrodes that show evidence of multiplexing are the ones whose RF straddles the two stimuli, because in that case, small eye movements will bring one of the two stimuli inside the RF.

2. Further analyses of eye position to rule out the possibility described above. In our discussions, it was noted that the new Figure 1 – Supplementary Figure 1 appears to show that numerous V1 RFs straddle the two stimuli, and under those conditions, we really want to know if the "multiplexed" responses are because of small fixational differences/microsaccades that affect which of the two stimuli takes a more dominant position in the RF. To test for this kind of effect, just the STD of eye position per trial for one-stimulus vs two-stimuli conditions does not seem to be sufficient. Instead, it seems important to know whether, in the two-stimuli condition, responses were more "A-like" when gaze put the A stimulus closer to the RF center, and were more "B-like" when gaze put the B stimulus closer to the RF center. So something like a linear regression of spiking response versus "eye position along the axis defined by the centers of the two stimuli, increasing towards the stimulus that alone elicited the larger response" could be useful.

*Reviewer #1 (Recommendations for the authors):*

The authors have made several changes in the manuscript to address previous concerns. However, the fact that typically only one stimulus spanned the RF makes it difficult to make a case for multiplexing, since the other stimulus is outside the classical RF. The arguments made by the authors in response to the previous RF-related question (point 2) are based on their own hypothesis about some underlying "spatial scale" of multiplexing which is coarser than the RF size of V1, which I do not find particularly convincing and would be extremely difficult to implement in V1. Further, while the authors showed more results related to eye position analysis, they do not show the key comparison that was requested previously.

To better appreciate the spatial scales involved, I refer to figures from the following two studies in V1 where very small stimuli (0.1 – 0.2 degrees) were used to map the RFs: Figure 2 of Xing et al., 2009, JNS, and Figure 2 of Dubey and Ray, 2016, JNP. Typically, the SD of fitted Gaussian is typically no more than 0.25 degrees at an eccentricity of 2-3 degrees (if you consider the radius as 2SD (0.5 degrees), the diameter is about a degree). For such RFs, there is no response if a stimulus is more than a degree away from the RF. For Gabor patches used in this paper, only the "size" is mentioned. Does size refer to the radius or SD? In either case, there is no way to fit two Gabors within the RF. What is the separation between two Gabors? Figure 1 should highlight all these details, including the radius (not just the center) of the V1 units.

Given the concern that one of the stimuli is always outside the RF, how do we explain the findings? One possible answer is in small differences in eye position. Multiplexing is anyway observed in only ~100 out of 1389 units. I suspect these are the units whose RF center is between the two stimuli. For the AB condition (i.e., both stimuli are presented), small jitters in eye position would bring one of the two stimuli in the RF, and therefore the unit would respond like either A or B. To address this, the authors should show the RF centers of the units that show evidence of multiplexing, along with the stimuli.

In addition, it is important to check for possible differences in eye position within AB conditions for trials for which responses were "A like" versus "B like". The authors have compared the AB condition to A and B conditions presented alone, but that is not enough. The argument that the stimuli were presented for a short duration and in pseudorandom order and therefore it is not possible for the animal to have systematically different eye positions for A/B versus AB conditions is obviously true, but that is not the point. The point is that the eye positions have small variations from trial to trial (as shown in Figure 1 – Supp 2) even before stimulus onset, and AB trials in which the position happens to be in one location get classified as "A like" and another location gets classified as "B like". It is in fact very hard to rule out this possibility given the instrument noise which affects the precision of eye positions, but it is crucial to rule this out.

*Reviewer #2 (Recommendations for the authors):*

The authors have appropriately addressed all comments. In particular, the controls on eye movements are sound.

*Reviewer #3 (Recommendations for the authors):*

The authors have addressed my concerns and all other concerns raised in the editor's summary. My main concern was whether uncontrolled fixational eye movements (microsaccades) could account in part for the observed multiplexing. I understand now that concern was largely because of missing details about eye position, RF sizes, stimulus sizes, etc, which are now reported. The possibility remains that trial-by-changes in eye position (within the fixation window) would inflate the proportion of single-neuron "mixture" cases for adjacent two-object stimuli (by effectively changing which object is inside the RF in any given trial). But, importantly, this could not explain the observed patterns of noise correlations.

[Editors' note: further revisions were suggested prior to acceptance, as described below.]

Thank you for resubmitting your work entitled "Coordinated multiplexing of information about separate objects in visual cortex" for further consideration by *eLife*. Your revised article has been evaluated by Joshua Gold (Senior Editor) and a Reviewing Editor.

The manuscript has been improved but there are some remaining issues that need to be addressed. We realized we were not as specific as we should have been in the last round of feedback and have tried to clarify here exactly what the reviewers are looking for.

In particular, because the authors claim that ~100 electrodes have mixture responses, the possible influence of eye movements should be tested for all the mixture units, not only 9/100 as done in the previous review. We would therefore like the analyses done for all the mixtures. To avoid ambiguity, we are listing the requested analyses in more detail below:

1. Show the RF centers of ALL the units labeled as mixtures (~100). The best way to do this would be to make a line passing through the centers of the two stimuli, and make five groups of units depending on their RF centers – (i) left of the left stimulus, (ii) on the left stimulus, (iii), between left and right stimulus, (iv) on the right stimulus and (v) right of the right stimulus. Then show what proportion of units in each category are mixtures. The eye movement hypothesis predicts that mixtures will be predominant in (iii). Actually, since 91/100 units have one of the two modes at zero (as far as I could understand the previous analysis), these units could even be at (i) or (v).

2. Do the eye position analysis for ALL mixture units. Since the stimuli are mainly separated along the x-axis, all we are asking is to make a scatter plot of spike counts and average eyeX position for each trial and then check for correlation between the two. This analysis is valid even for units that only respond to one stimulus (i.e. the remaining 91 units). We expect to find a significant correlation (either positive or negative) if the results are due to small eye movements, but not if the firing is due to their multiplexing hypothesis. Or, if you find significant correlations since the RF sizes are comparable to the stimulus sizes, you need to show that these correlations are insufficient to explain the bimodality.

3. In addition, I think the authors should at least show the typical RF radius of the units (make a circle of radius of 0.5 degrees on a few of the units, perhaps the ones shown in Figure 4). We think it is important to show that the RFs do not encompass both stimuli, which is not clear from the plots.

4. The fact that one mode of the bimodal distribution is zero in 91/100 cases should also be made clearer, perhaps in the methods section. Essentially the bimodal response in most cases is not A versus B, but A/B versus zero.

---

## [Author Response]

Essential revisions:While this paper shows some intriguing results, we feel that there are a lot of open questions that need to be addressed before convincing evidence of multiplexing can be established. These points are discussed below:1. The best spike count model shown in Figure 2C is confusing. It seems that the number of "conditions" is a small fraction of the total number of conditions (and neurons?) that were tested. Supplementary Figure 1 provides more details (for example, the "mixture" corresponds to only 14% of total cases), but it is still confusing (for example, what does WinProb>Min mean?). From what I understood, the total number of neurons recorded for the Adjacent case in V1 is 1604, out of which 935 are Poisson-like with substantially separated means. Each one has 2 conditions (for the two directions), leading to 1870 conditions (perhaps a few less in case both conditions were not available). I think the authors should show 5 bar plots – the first one showing the fraction for which none of the models won by 2/3 probability, and then the remaining 4 ones. That way it is clear how many of the total cases show the "multiplexing" effect. I also think that it would be good to only consider neurons/conditions for which at least some minimum number of trials are available (a cutoff of say ~15) since the whole point is about finding a bimodal distribution for which enough trials are needed.

We thank you for catching this confusion. We have revised Figure 2C as requested, modified Supplementary Figure 1 (now Figure 2 – Supp Figure 1) to be more clear, and improved Table 1 and its legend to more succinctly walk the reader through the numbers of included conditions and how many units and distinct sessions they concern. We have also added a Table 2 to provide the trial counts, which exceed the criteria we have previously benchmarked and used (Mohl et al 2020; Caruso et al., 2018).

Details:

We first explain the changes to Table 1. The most important column is the last one; this column provides the total numbers of conditions (“Triplets”) that passed the Poisson and good response-separation criteria for inclusion in the analyses. The remaining columns illustrate how many units and sessions these included triplets came from. For example, in the adjacent data set in V1, there were 16 sessions; all 16 sessions yielded data that was included. There were 1604 units of which 935 yielded data that was included. We’ve re-written the legend to better explain this.

Note: if the reviewers feel it would be beneficial to simplify, we could easily delete the “Available sessions” and “Available units” columns, and just report the number of sessions and units that were actually used.

Regarding “the total number of neurons recorded for the Adjacent case in V1 is 1604, out of which 935 are Poisson-like with substantially separated means. Each one has 2 conditions (for the two directions), leading to 1870 conditions” – this is not quite correct, but it was a very helpful explanation of how we had confused the matter. To summarize the Adjacent case in V1, 935 units provided triplets, 1-2 each, yielding a total of 1389.

For Figure 2C (now 2C and D), “I think the authors should show 5 bar plots - the first one showing the fraction for which none of the models won by 2/3 probability, and then the remaining 4 ones.”, we have included a pie chart to complement the bar chart to illustrate the total number of conditions involved.

For Supplementary Figure 1 *“what does WinProb>Min mean”,* that refers to the lowest possible winning probability. We used a flat prior, i.e. each model starts off with a 0.25 probability of being the best model. We have modified this figure (now Figure 2 – Supplementary Figure 1) to replace “WinProb>Min” with “WinProb>0.25”.

Regarding “I also think that it would be good to only consider neurons/conditions for which at least some minimum number of trials are available (a cutoff of say ~15) since the whole point is about finding a bimodal distribution for which enough trials are needed”, we think we are in good shape on this front, but we agree we had not documented this. Accordingly, we now include Table 2 which is accompanied by following text in the Methods section: “The numbers of trials involved for the different data sets are provided in Table 2. The trial counts were adequate to provide accurate model identification according to our previous simulations. Depending on the separation between λA and λB, we previously found that model identification accuracy in simulations is high for trial counts as low as 5 “AB” trials, and plateaus near ceiling around “AB” trial counts of about 10 trials and above – i.e. below the mean trial counts available here for all datasets (see Figure 4 of Mohl et al., 2020).”

The relevant figure supporting this point is Figure 4 from Mohl et al. (2020).

Mohl JT, Caruso VC, Tokdar S, Groh JM (2020) Sensitivity and specificity of a Bayesian single trial analysis for time varying neural signals. Neurons, Theory, Data Analysis, and Behavior.

2. More RF details need to be provided. What was the size of the V1 RFs? What was the eccentricity? Typically, the RF diameter in V1 at an eccentricity of ~3 degrees is no more than 1 degree. It is not enough to put 2 Gabors of size 1 degree each to fit inside the RF. How close were the Gabors? We are confused about the statement in the second paragraph of page 9 "typically only one of the two adjacent gratings was located within the RF" – we thought the whole point of multiplexing is that when both stimuli (A and B) are within the RF, the neuron nonetheless fires like A or B? The analysis should only be conducted for neurons for which both stimuli are inside the RF. When studying noise correlations, only pairs that have overlapping RFs such as both A and B and within the RFs of both neurons should be considered. The cortical magnification factor at ~3-degree eccentricity is 2-2.5mm/degree, so we expect the RF center to shift by at least 2 degrees from one end of the array to the other.

We now provide the requested details regarding stimulus size and receptive field locations in the Methods. But before summarizing those results, we would like to clarify a conceptual point: while our study is motivated by the observation that tuning in sensory structures is coarse relative to our perceptual capabilities, we did not mean to limit this consideration to the spatial scale of V1 neurons specifically. Rather, we see this as an overall problem across sensory pathways. For example, even if two particular stimuli activate completely separate populations in V1, this may not be the case at later stages of processing such as IT cortex where receptive fields are quite large. We believe the brain has to solve this problem at the scale of the entire sensory processing stream, and It must do so without prior knowledge of the particular spacing of particular individual stimuli.

That said, we agree that it could well have turned out that fluctuating activity might only be evident when two stimuli were within the same receptive fields of individual neurons in a particular brain area. However, this did not turn out to be the case. Many of the units in the present study only had one stimulus in the receptive field, yet a subset showed evidence of fluctuating response patterns.

Perhaps we would have observed *more* evidence for multiplexing if both stimuli were in the receptive field. Indeed, in our previous study involving the inferior colliculus and inferotemporal cortex (Caruso et al 2018), we observed a higher incidence of multiplexing than in the current study. In the IC, spatial tuning is very broad (sound location is rate coded in the mammalian brain), so in general most neurons responded to both sounds. We are currently working on a followup study in which we are parametrically varying the frequency of the two sounds to smoothly vary the response differences between the A and B stimuli. In this new study, we find that multiplexing is indeed more prevalent when the two stimuli are both in the frequency response area of the neuron than when only one stimulus is. So the fact that in general only one stimulus drove the units in the current study may well have limited the evidence for multiplexing in the current dataset.

Changes made: We have modified the language in the introduction to describe the rationale more carefully, so as not to imply that multiplexing need only occur in cases where the two stimuli are within the same receptive field of the neurons under study in a particular brain area. Rather, multiplexing is likely to have some as yet unknown characteristic spatial scale that may be determined by the coarsest tuning evident at any stage in the sensory pathway. We also now return to this topic in the Discussion to specifically clarify our interpretation regarding this issue. We have future experiments planned in which we will systematically vary the locations of the stimuli involved, so we expect to have data to address this issue in the future.

Key changes in the introduction: The sentence “Such breadth of tuning means that individual neurons will commonly experience more than one stimulus to which they could in principle respond, making it unclear how information about multiple objects is preserved.” has been rewritten as: “Such breadth of tuning means that there will be overlap in the population of neurons activated by individual stimuli, making it unclear how information about multiple objects is preserved.” This shifts the meaning to focus on the (implied) population overlap rather than the receptive fields of the particular individual neurons we studied.

Key changes in the Methods: The requested information regarding receptive fields and stimulus locations is provided in the Methods section under “Electrophysiological recordings and visual stimuli”: “…we analyzed trials in which attention was directed to a Gabor patch located in the hemisphere ipsilateral to the recorded V1 neurons (i.e. well away from those neurons’ receptive fields, see below). On these trials, two unattended Gabor patches were presented in close proximity to each other within the area covered by the receptive fields of the recorded V1 neurons – these receptive fields were approximately 3 degrees eccentric and had classical receptive field diameters typically estimated to be <1 degree of visual angle. These patches were centered 2.5-3.5 degrees eccentrically and each stimulus typically subtended 1 degree of visual angle (see Ruff and Cohen 2016 Figure 1B for a sketch, reproduced here in Figure 1 – Supplementary Figure 1)…”

Key changes in the Discussion: We inserted a new paragraph that reads “It is interesting to note that we observed evidence of multiplexing each stimulus even in V1 where there receptive fields are small and the stimuli we used did not themselves typically span more than one receptive field – thus the coarseness of tuning did not necessarily pose a problem for the encoding of these particular stimuli in this particular brain area. However, the precision of V1’s spatial code may not be the limiting factor.

Multiplexing is likely to have some as yet unknown characteristic spatial scale that may be determined by the coarsest tuning evident at any stage in the sensory pathway. Future work in which stimuli are systematically varied to manipulate the amount of overlap in the activity patterns evoked in different brain areas by each stimulus alone are needed to answer this question. “

3. Eye data analysis: the reviewers are concerned that this could potentially be a big confound. Removing trials that had microsaccades is not enough. Typically, in these tasks the fixation window is 1.5-2 degrees, so that if the monkey fixates on one corner in some trials and another corner in other trials (without making any microsaccades in either), the stimuli may nonetheless fall inside or away from the RFs, leading to differences in responses. This needs to be ruled out. We do not find the argument presented on pages 18 or 23 completely convincing, since the eye positions could be different for a single stimulus versus when both stimuli are presented. It is important to show that the eye positions are similar in "AB" trials for which the responses are "A" like versus "B" like, and these, in turn, are similar to when "A" and "B" are presented alone.Relatedly, more details on the detection of microsaccades and threshold values for inclusion (relative to stimulus and RF sizes) should be provided, given how central a role they might play. In particular, the authors state in Discussion that small residual eye movements would inflate response variability in all stimulus conditions. This is correct, but because of the stimulus design, it is possible (likely?) that the effects are quite different for segregated stimuli versus superimposed and single-stimulus conditions. Furthermore, the difference might be precisely in the direction of the effects reported. That is because segregated stimuli are spatially separate, and each stimulus only covers some receptive fields in the recording, it is possible that eye movements would bring inside the RF a different stimulus in each trial. In addition to producing bimodal response distributions for individual neurons, this would also induce positive correlations for pairs with the same preference and negative correlations for pairs with opposite preferences. On the other hand, in the superimposed condition where the stimulus is large enough to cover all RFs, at most eye movements would bring (part of) the stimulus inside versus outside the RF across trials, therefore contributing to positive noise correlations for all pairs (i.e., to shifts from stimulus-driven to spontaneous activity, in the extreme case). In our opinion, the main concern raised in the public review deserves more in-depth analysis, of the data and/or simulations, for us to be convinced that eye movements truly do not play a role. Conversely, if they do, it may be worth discussing the possibility that they are part of the mechanism underlying the proposed coding scheme.

We appreciate the reviewers raising these concerns, which we now address more fully in the manuscript with a combination of new data analysis, additional methodological details, and more fully fleshed-out reasoning. The key points are as follows:

(a) Eye position did not differ during the A/B vs. AB stimulus presentations. See new Figure 1 -Supplementary Figure 2 in the manuscript, and included below. This is the simplest answer to the above questions, but lest any concerns remain, read on:

(b) The timing doesn’t work for variation in eye position to pose a substantial confound. The A/B/AB stimuli are delivered in pseudorandom order, so it was not possible for the animals to anticipate which stimulus would appear; thus pre-stimulus eye position effects are not a possibility. And there isn’t time for post-stimulus eye position effects to have a meaningful influence during the short spike counting window we used – 200 ms after stimulus onset. It typically takes 150-350 ms for visual stimuli to elicit a macrosaccade; a similar latency is likely involved for any potential stimulus-influenced modulation of fixational eye movements (Engbert and Kliegl, 2003). Because we only analyzed a 200 ms spike counting temporal epoch, even if stimuli were to differentially modulate the position of the eyes within the fixation window, there is very little time for such modulation to affect the spikes we counted.

(Note – any saccades directed to one or the other targets would of course be excluded as incorrect trials, but we understood the concern to relate to any more subtle tendencies to *fixate* differently based on the stimulus conditions.)

(c) Given that there was no difference in eye position between conditions in the adjacent stimuli data sets, we believe it is beyond the scope of this paper to conduct this analysis for the superimposed stimuli datasets as the lack of an effect in the adjacent dataset is sufficient to eliminate this as a candidate explanation for our findings.

(d) Details regarding microsaccade detection, RF sizes, stimulus sizes, and fixation windows are now included as described above. We apologize for omitting these details. To recap for the adjacent V1 dataset:

Fixation windows were +/- 0.5 degrees.

RF sizes: typically less than 1 degree

Stimulus sizes: typically 1 degree, about 2.5-3.5 eccentric

Microsaccade detection criteria: stimulus presentations in which eye velocity ever exceed six standard deviations from the mean eye velocity during fixation were excluded from further analysis, following the method of Engbert and Kliegl, 2003.

(f) Regarding “this would also induce positive correlations for pairs with the same preference and negative correlations for pairs with opposite preferences”, we respectfully disagree. This would only be the case if the receptive fields of the neurons with the same preferences were truly precisely aligned in space, as opposed to merely overlapping. As shown in Ruff and Cohen 2016 Figure 1B, and now illustrated in our Figure 1 – Supplementary Figure 1, this was not the case. The logic is most easily conveyed in Author response image 1:

**Author response image 1. sa2fig1:** This schematic focuses on the “congruent” situation, assuming these involve neurons with adjacentreceptive fields. Even within this group, variation in eye position would create both positive and negative correlations, and would not be expected to selectively impact dual stimulus trials or congruent vs incongruent pairs of neurons.

4. Figures 5 and 6 show that the difference in noise correlations between the same preference and different preference neurons remains even for non-mixture type neurons. So, although the reason for the particular type of noise correlation was given for multiplexing neurons (Figure 3 and 4), it seems that the same pattern holds even for non-multiplexers. Although the absolute values are somewhat different across categories, one confound that still remains is that the noise correlations are typically dependent on signal correlation, but here the signal correlation is not computed (only responses to 2 stimuli are available). If there is any tuning data available for these recordings, it would be great to look at the noise correlations as a function of signal correlations for these different pairs. Another analysis of interest would be to check whether the difference in the noise correlation for simply "A"/"B" versus "AB" varies according to neuron pair category. Finally, since the authors mention in the Discussion that "correlations did not depend on whether the two units preferred the same stimulus or different", it would be nice to explicitly show that in figure 5C by showing the orange trace ("A" alone or "B" alone) for both same (green) and different (brown) pairs separately.

We thank you for this interesting suggestion. We have conducted the desired analysis (see Author response table 1). The basic conclusions have not changed as described below.

**Author response table 1. sa2table1:** Median spike count correlations for additional subgroups of the V1 adjacent stimuli dataset. The top two rows show the median spike count correlations observed for dual stimuli for various types of pairs of units, and correspond to the data shown in figures 4 and 5 in the main text (first two rows). The next three rows show the same analyses conducted for trials involving single stimuli. Here, the “congruent” group was subdivided according to whether the presented stimulus was the one that elicited the stronger response (“driven”) or the weaker one (“not driven”). The bottom two rows show the differences in the medians observed for the relevant congruent and incongruent groups (lines 1 minus 2 and lines 3 minus 5)

	All	Mixture-Mixture pairs	Intermediate-Intermediate pairs	Single-Single pairs
Congruent AB	0.252	0.486	0.263	0.233
Incongruent AB	-0.052	-0.140	-0.009	-0.032
Congruent driven A or B	0.251	0.384	0.266	0.250
Congruent not driven A or B	0.145	0.132	0.142	0.147
Incongruent A or B	0.116	0.127	0.076	0.108
				
Congruent minus Incongruent AB	0.304	0.626	0.272	0.265
Congruent driven minus incongruent A or B	0.135	0.257	0.190	0.142

Details: This analysis required one additional complexity, namely that the concepts of “congruent” and “incongruent” don’t translate cleanly to the single stimulus case. Consider an “incongruent” pair in which unit 1 prefers “A” and unit 2 prefers “B”. When “A” is presented, unit 1 will be strongly driven but unit 2 will not. When “B” is presented, the pattern will be the opposite. These two cases are thus more or less equivalent. However, for “congruent” pairs, when they both prefer “A” and “A” is the stimulus that is presented, the correlations could be quite different than when “B” is presented. Accordingly, for the “congruent” pairs, we subdivided according to whether the stimuli were the better (“driven”) or worse (“not driven”) of the two stimuli, and we focused the comparison on the difference between the congruent-driven and the incongruent spike count correlations. The bottom two rows of the Supplementary Table show the results for the single stimulus (A or B) conditions compared to the dual stimulus (AB) conditions.

Results: The basic pattern of a larger difference in the correlation patterns between the Congruent and Incongruent units for dual (AB) stimuli than for individual stimuli (A or B) holds across all the different categories of tested pairs. Compare the bottom two rows:

We have revised the wording in the Discussion to be consistent with the data as presented.

5. We are confused about the nature of Poisson models. If we are correct, the Poisson(a+b) is the sum of the two Poisson(a) and Poisson(b), that is, Poisson(a+b) = Poisson(a) + Poisson(b). Then, the mixture and intermediate models are very similar, identical if a*λ_A and (1-a)*λ_B happen to be integer numbers.

We thank the reviewers for bringing this confusion to our attention. We have modified Figure 2 to be more clear. The key distinction involves the rate parameter (λ) vs the draws from the Poisson distribution involving that rate parameter. This is most easily explained with an expanded illustration in Author response image 2:

**Author response image 2. sa2fig2:** The red and blue curves illustrate two Poisson distributions with rates A and B. The black curve illustrates draws from a mixture of those two poissons – in this case, there is a 50% chance that the draw is from rate A and a 50% chance from rate B. The green curve illustrates draws from a single Poisson whose rate is equal to A+B. We would classify the green curve as an outside because A+B>max(A,B). An intermediate is not shown here but it would have a shape like the green curve and a peak between the red and blue curves – e.g. Poi((A+B)/2).

These patterns (and more) are illustrated in Figure 2B, and we have clarified the associated equations to be readable in plain language – see the section now entitled “Spike count drawn from:” on the right hand side:

6. It is unclear why the 'outside' model predicts responses outside the range if neurons were to linearly sum the A and B responses.

We hope this is now more clear with the improved explanation above. Suppose a neuron’s average response to a stimulus “A” is 5 spikes and a stimulus “B” is 10 spikes. If “A” and “B” are both presented together and the neuron responds with 15 spikes, this is linear summation. We would define it as an “outside” response because 15 is not between 5 and 10. In short, linear summation is very rare in the V1 adjacent dataset, since we almost never observed “outside” classifications.

7. It is also unclear why the 'single' hypothesis would indicate a winner-take-all response. If we understand correctly, under this model, the response to A+B is either the rate A or B, but not the max between λ_A and λ_B. Also, this model could have given an extra free parameter to modulate its amplitude to the stimulus A+B.

Correct – “single” can also be “loser-take-all”. This is now stated on page 9 (“indicating a winner (or loser) -take-all response patterns”). Regarding an extra free parameter related to amplitude (of response?), we are not sure we follow. The distributions of spike counts observed over the set of trials involving the A, B, and AB stimuli are fully incorporated into the model comparison. Thus, response “amplitude”, meaning spike count, is already evaluated by the model.

8. The concept of "coarse population coding" can be misleading, as actual population coding can represent stimulus with quite good precision. The authors refer to the broad tuning of single cells, but this does not readily correspond to coarse population coding. This could be clarified.

We thank the reviewers for noting that clarification is needed on this issue. Our key point is that while codes of coarsely tuned units can produce very precise estimates of the value being encoded - c.f. excellent modeling work by Baldi and Heiligenberg (1988) – this has only been shown in circumstances where only one value is being encoded, such as in motor systems. We can only make one movement with a given body part at a given moment in time, whereas sensory systems must at least initially encode a broad array of stimuli.

We have revised the wording of the introductory paragraph to provide more emphasis on the “one movement” limitation of previous work involving coarse coding in motor systems (Introduction, paragraph 1: “However, individual motor systems only generate one movement at a time.”).Baldi P, Heiligenberg W (1988) How sensory maps could enhance resolution through ordered arrangements of broadly tuned receivers. Biological Cybernetics 59:313-318.

9. As a complement to the correlation analysis, one could check whether, on a trial-by-trial basis, the neuronal response of a single neuron is closer to the A+B response average, or to either the A or B responses. This would clearly indicate that the response fluctuates between representing A or B, or simultaneously represents A+B. I am trying to understand why this is not one of the main analyses of the paper instead of the correlation analysis, which involves two neurons instead of one.

We thank the reviewers for this suggestion and have added a new figure (Figure 9) to the manuscript. Details: At the individual cell level, the analysis suggested above is accomplished formally by the Bayesian model comparison presented in Figure 2 and subsequent figures – units classified as “mixtures” have responses on a trial-by-trial basis that alternate between A-like and B-like patterns.

Then, the coordinations in these fluctuations are assessed using the pair-wise correlation analysis.

What we now add is an example recording session showing the full pattern of fluctuations across a set of simultaneously recorded “mixture”-responding units. This example shows all of the main findings - individual units alternating between A- and B-like responses across trials, positive correlations between some pairs of units, negative correlation between others, and lack of correlation between yet others.

There is a bias in the overall representation – in this case, “A” is over represented compared to “B”.

However, on any given individual trial, there are some units responding in an “A-like” fashion and others in a “B-like” fashion (red and blue squares are present in every column).

10. In the discussion about noise correlations, the recent papers Nogueira et al., J Neuroscience, 2020 and Kafashan et al., Nat Comm, 2021 could be cited. Also, noise correlations can also be made time-dependent, so the distinction between the temporal correlation hypotheses and noise correlations might not be fundamental.

We appreciate having these excellent studies brought to our attention, and have included them in the list of citations. We concur that the temporal correlation and noise correlation distinctions may be a matter of techniques employed to date.

11. It would be interesting to study the effect of contrast on the mixed responses. Is it reasonable to predict that with higher contrast the mixture responses would be more dominant than the single ones? This could be the case if the selection mechanism has a harder time suppressing one of the object responses. This would also predict that noise correlations will go down with higher contrast.

This would be interesting but we do not have the data to address this question for this particular study. In a study that is currently in progress in the Groh lab, we have evidence in the inferior colliculus that when a visual stimulus is paired with one of two sounds, the neural responses are biased to the visually-paired sound in comparison with the non-visually-paired sound. This will be its own separate publication when we have completed the data collection in a second animal.

12. What is the time bin size used for the analysis? Would the results be the same if one focuses on the early time responses or on the late time responses? At least from the units shown in Figure 2, it looks that there is always an object response that is delayed respect to the other, so it would seem interesting to test noise correlations in those two temporal windows.

We apologize for the confusion, but the only time bin used in the current study is the 200-ms spike counting window. The examples shown in Figure 2 are spike count distribution plots, not PSTHs - the x axis is spike counts, not time. We have revised the title of Figure 2A to make this more clear.

[Editors' note: further revisions were suggested prior to acceptance, as described below.]

The manuscript has been improved and many of the reviewers' concerns have been addressed. However, some major issues remain. Although we typically avoid repeated revise/resubmit cycles, we believe that it is important for these issues to be addressed in a new revision. Specifically, upon discussion the reviewers unanimously remained concerned about the possibility that at least some of your results could be accounted for by subtle differences in eye movements. The new analyses related to that issue were appreciated but considered inadequate.<break />As detailed below, we would like you to provide:<break />1. More information about whether the electrodes that show evidence of multiplexing are the ones whose RF straddles the two stimuli, because in that case, small eye movements will bring one of the two stimuli inside the RF.

Among the 100 triplets classified as showing “mixture” response patterns with a winning probability of at least 0.67 (i.e. “mixture” is 2X as likely as all other possible classifications combined), we identified only 9 triplets exhibiting responses to both the A and B stimuli alone, as defined as a response greater than 1 SD above baseline.

As context, recall that one of the critical exclusionary screens for all the analyses in this paper is that a unit’s responses to the A and B stimuli must be sufficiently different that the Bayesian spike count distribution classification can be conducted. If the responses are too similar, it is not possible to evaluate whether a given spike count is more “A-like” or “B-like” etc. So, it is perhaps not very surprising that so few of our “mixtures” were responsive to both “A” and “B” stimuli.

Nevertheless, we proceeded with the additional analysis suggested below.

2. Further analyses of eye position to rule out the possibility described above. In our discussions, it was noted that the new Figure 1 – Supplementary Figure 1 appears to show that numerous V1 RFs straddle the two stimuli, and under those conditions, we really want to know if the "multiplexed" responses are because of small fixational differences/microsaccades that affect which of the two stimuli takes a more dominant position in the RF. To test for this kind of effect, just the STD of eye position per trial for one-stimulus vs two-stimuli conditions does not seem to be sufficient. Instead, it seems important to know whether, in the two-stimuli condition, responses were more "A-like" when gaze put the A stimulus closer to the RF center, and were more "B-like" when gaze put the B stimulus closer to the RF center. So something like a linear regression of spiking response versus "eye position along the axis defined by the centers of the two stimuli, increasing towards the stimulus that alone elicited the larger response" could be useful.

Of these 9 doubly-responsive triplet conditions identified above, only 1 showed a strongly statistically significant effect of fixational variation on the response patterns (linear regression as requested above, i.e. spiking response as a function of eye position along the axis defined by the centers of the two stimuli, p<0.01; two others showed borderline effects 0.05>p>0.025 whose potential significance did not survive when checked in a second multiple linear regression with both parallel and orthogonal dimensions of eye position included).

In short, the number of triplets involving any provable impact of eye position on the response patterns is a very small proportion of the total and does not impact the overall pattern of the results.

The outcome of these control experiments is summarized here:

**Author response image 3. sa2fig3:** 

Reviewer #1 (Recommendations for the authors):The authors have made several changes in the manuscript to address previous concerns. However, the fact that typically only one stimulus spanned the RF makes it difficult to make a case for multiplexing, since the other stimulus is outside the classical RF. The arguments made by the authors in response to the previous RF-related question (point 2) are based on their own hypothesis about some underlying "spatial scale" of multiplexing which is coarser than the RF size of V1, which I do not find particularly convincing and would be extremely difficult to implement in V1. Further, while the authors showed more results related to eye position analysis, they do not show the key comparison that was requested previously.

It is true that the hypothesis about a coarser-than-V1-RF spatial scale is only a hypothesis, but it is consistent with the observations of our paper so we feel it is a reasonable speculation to offer. We hope future studies will be able to answer this question. The underlying mechanism(s) that might create these multiplexed signals – whether in V1 or elsewhere – likewise remains unknown.

To better appreciate the spatial scales involved, I refer to figures from the following two studies in V1 where very small stimuli (0.1 – 0.2 degrees) were used to map the RFs: Figure 2 of Xing et al., 2009, JNS, and Figure 2 of Dubey and Ray, 2016, JNP. Typically, the SD of fitted Gaussian is typically no more than 0.25 degrees at an eccentricity of 2-3 degrees (if you consider the radius as 2SD (0.5 degrees), the diameter is about a degree). For such RFs, there is no response if a stimulus is more than a degree away from the RF. For Gabor patches used in this paper, only the "size" is mentioned. Does size refer to the radius or SD? In either case, there is no way to fit two Gabors within the RF. What is the separation between two Gabors? Figure 1 should highlight all these details, including the radius (not just the center) of the V1 units.

1. We thank the reviewer for pointing out these omissions. We have clarified the text associated with Figure 1, and we now include the stimulus location details in an expanded Figure 1 Supplementary Figure 1, which has a new panel B:

In the main text, the legend to Figure 1 now clarifies the size/diameter issue and points explicitly to Figure 1 Supplementary Figure 1 for the full description:

“C. In the “superimposed” dataset, gratings were presented either individually or in combination at a consistent location and were large enough to cover the V1 and V4 receptive fields (stimulus diameter range: 2.5-7o,). The combined gratings appeared as a plaid (rightmost panel). Monkeys maintained fixation throughout stimulus presentation and performed no other task. D. In the V1 “adjacent” dataset, Gabor patches were smaller (typically ~1o,see Figure 1 Supplementary Figure 1).”

As for the receptive fields, these were mapped with stimuli of fixed sizes, permitting identification of the centers as shown in Supplementary Figure 1A, but the precise boundaries were not systematically evaluated as this would require varying the sizes of the mapping stimuli. This limitation reflects the original experimental needs of choosing satisfactory stimulus locations for performance of the behavioral task and optimizing the stimulus positions for the simultaneously recorded MT units (Ruff and Cohen 2016; MT data not analyzed here). Hence, we must rely on a rough sense inferred from a combination of the stimulus positions and the inclusion criteria: i.e. only units responsive to at least one of these stimuli were included for analysis.

2. We appreciate the elegant work of Xing et al., 2009 and Dubey and Ray, 2016 on receptive field mapping and have included citations to these studies.

Given the concern that one of the stimuli is always outside the RF, how do we explain the findings? One possible answer is in small differences in eye position. Multiplexing is anyway observed in only ~100 out of 1389 units. I suspect these are the units whose RF center is between the two stimuli. For the AB condition (i.e., both stimuli are presented), small jitters in eye position would bring one of the two stimuli in the RF, and therefore the unit would respond like either A or B. To address this, the authors should show the RF centers of the units that show evidence of multiplexing, along with the stimuli.

See analysis above. Only 9 (9%) of the “mixture”-classified units are receptive field straddlers. Note that to be included for analysis, units had to exhibit different responses to the A and B stimuli individually; this tended to limit the number of units responsive to both that were included in the final tally.

In addition, it is important to check for possible differences in eye position within AB conditions for trials for which responses were "A like" versus "B like". The authors have compared the AB condition to A and B conditions presented alone, but that is not enough. The argument that the stimuli were presented for a short duration and in pseudorandom order and therefore it is not possible for the animal to have systematically different eye positions for A/B versus AB conditions is obviously true, but that is not the point. The point is that the eye positions have small variations from trial to trial (as shown in Figure 1 – Supp 2) even before stimulus onset, and AB trials in which the position happens to be in one location get classified as "A like" and another location gets classified as "B like". It is in fact very hard to rule out this possibility given the instrument noise which affects the precision of eye positions, but it is crucial to rule this out.

We apologized for the previous confusion: we understood the previous concern to involve the very distinct patterns of spike count correlation exhibited on dual stimulus trials rather than the mixture classification itself. We hope the reviewer’s concern is now satisfactorily addressed by the analysis described above.

Reviewer #3 (Recommendations for the authors):The authors have addressed my concerns and all other concerns raised in the editor's summary. My main concern was whether uncontrolled fixational eye movements (microsaccades) could account in part for the observed multiplexing. I understand now that concern was largely because of missing details about eye position, RF sizes, stimulus sizes, etc, which are now reported. The possibility remains that trial-by-changes in eye position (within the fixation window) would inflate the proportion of single-neuron "mixture" cases for adjacent two-object stimuli (by effectively changing which object is inside the RF in any given trial). But, importantly, this could not explain the observed patterns of noise correlations.

We thank you, and see above analyses for reassurance on the mixture-inflation question.

[Editors' note: further revisions were suggested prior to acceptance, as described below.]

The manuscript has been improved but there are some remaining issues that need to be addressed. We realized we were not as specific as we should have been in the last round of feedback and have tried to clarify here exactly what the reviewers are looking for.

We appreciate the efforts of the reviewers to help us improve our paper and make it as strong and clear as possible. Before we delve into the detailed results of the newly requested analyses – which we think the reviewers will find reassuring – we think it is worth re-capping the evidence as a whole regarding the main topic at issue, which we understand to be eye position/eye movement as a possible factor in the observations reported in this paper and the inferences we draw from those observations.

The observations in this paper fall into two overarching categories: (1) a “multiplexing” analysis, first introduced in Caruso et al. 2018 (involving recordings in an auditory brain area (the inferior colliculus) and a visual brain area (the face patch system of IT cortex); and (2) patterns of spike count (“noise”) correlations between pairs of simultaneously recorded neurons.

Our present paper shows that the pattern of spike count correlations in V1 is very different when two stimuli are presented vs when only one stimulus is presented – there are stronger correlations, in both positive and negative directions, than observed when only one stimulus is presented (Figure 4). Regarding contributions of eye position to this pattern, as discussed in earlier rounds or review, the single- and dual-stimulus conditions were randomly interleaved and trials with microsaccades excluded, so variation in eye position cannot account for this aspect of our findings (Figure 1 Supplementary Figure 2). Our understanding is that the reviewers are satisfied as to this point.

In the immediately prior and current rounds of review, concerns have centered further on the potential role of eye position in the “mixture” classification itself. For this to undermine our conclusions, such an effect would need to be sufficiently prevalent that any remaining results after excluding potentially confounded ones are insufficient to support the overall interpretations. Put simply, for correlation-witheye-position to be a confound, it would have to be some combination of *big*, *common*, *systematic*, and *related not only to the mixture classification but also the spike count correlation*. We think correlation with-eye-position fails as an explanation due to shortcomings in all of these realms.

First: how common is correlation with eye position? In the previous round, we were asked to consider a subset of the “mixtures”, those that were doubly-responsive to both stimuli. We showed that only about 11% of this subset had a significant effect of eye position evident in the response patterns. In the current round, reviewers requested that we extend this analysis to the full population of mixtures, which we have now done (detailed below). Again, only about 9% of the conditions labeled “mixture” exhibited a significant correlation with eye position. This compares with about 5% of the conditions not labeled as “mixtures” showing such an effect. The prevalence of eye position correlation in “mixtures” vs “non mixtures” is not significantly different by chi-square analysis (p=0.17).

Second: is correlation with eye position systematically related to stimulus location/receptive field location? The analyses requested by the reviewers (and presented in full below) answer this question definitively: the incidence of a correlation with eye position is not related to the relationship between RF location and stimulus location in any particularly obvious way.

Third: If we were to exclude any of the modest number of conditions that showed a correlation with eye position, the remaining observations in the manuscript would not change in any meaningful way. Indeed, our manuscript already shows that while the bimodal spike count correlation pattern is especially strong when both units in a pair are classified as “mixtures” (Figure 4) it is *also seen other combinations of response patterns* (Figure 5). This supports the interpretation that even if variation in eye position were to erroneously contribute to labeling of some conditions as “mixtures” when they shouldn’t be, this error would not meaningfully undermine the overall observations regarding spike count correlations differing on single vs dual stimulus conditions.

Zooming out to a larger context, as noted above, we have previously identified “mixture” response patterns in an auditory brain region and a visual brain region with much larger receptive fields than those in V1 (Caruso et al. 2018). Eye position within a small fixation window is unlikely to have affected the classification of response patterns in these areas. We also have numerous additional datasets (not yet published) that support “mixtures” as a general phenomenon. For example, we have seen “mixture” response pattern in datasets involving visual and auditory stimuli and recordings in the superior colliculus and prefrontal cortex, as well as additional data involving visual stimuli and area MT. Thus, we are quite confident that “mixtures” do not reflect some weird artifact or exceptional sensitivity to eye position that might hold true only for the present study.

We turn below to the specific analyses requested:

In particular, because the authors claim that ~100 electrodes have mixture responses, the possible influence of eye movements should be tested for all the mixture units, not only 9/100 as done in the previous review. We would therefore like the analyses done for all the mixtures. To avoid ambiguity, we are listing the requested analyses in more detail below:1. Show the RF centers of ALL the units labeled as mixtures (~100). The best way to do this would be to make a line passing through the centers of the two stimuli, and make five groups of units depending on their RF centers – (i) left of the left stimulus, (ii) on the left stimulus, (iii), between left and right stimulus, (iv) on the right stimulus and (v) right of the right stimulus. Then show what proportion of units in each category are mixtures. The eye movement hypothesis predicts that mixtures will be predominant in (iii). Actually, since 91/100 units have one of the two modes at zero (as far as I could understand the previous analysis), these units could even be at (i) or (v).2. Do the eye position analysis for ALL mixture units. Since the stimuli are mainly separated along the x-axis, all we are asking is to make a scatter plot of spike counts and average eyeX position for each trial and then check for correlation between the two. This analysis is valid even for units that only respond to one stimulus (i.e. the remaining 91 units). We expect to find a significant correlation (either positive or negative) if the results are due to small eye movements, but not if the firing is due to their multiplexing hypothesis. Or, if you find significant correlations since the RF sizes are comparable to the stimulus sizes, you need to show that these correlations are insufficient to explain the bimodality.

As requested in point #1, Figure 1—figure supplement 3 shows the RF centers of all* the units, aligned on one of the two stimuli for clarity (since the relationship between the two stimuli was relatively consistent, this is a reasonable simplification that allows for visualization). The data points indicate the RF centers of conditions labeled as mixtures (pink dots) and those labeled as non-mixtures (green dots); conditions that passed criterion for inclusion but did not achieve a high-confidence label in the spike-count distribution analysis are shown in yellow-orange.

Breaking these down into the groups as requested in the second part of point #1 above, we get:

**Author response table 2. sa2table2:** 

RF location	Left of left stim (group “i”)	“In” left stim(group “ii”)	Between stims(group “iii”)	“In” right stim(group “iv”)	Right of right stim (group “v”)
totals	37	128	58	28	0
“mixtures”	22	45	19	4	0
Percent “mixtures”	59%	35%	33%	14%	-
Num mixtures also eye position sensitive	3	4	2	0	-
Percent of mixtures that are eye position sensitive	13.6%	8.9%	10.5%	0%	-

Note that following the method requested by the reviewers in point #1, the “in” groups also include units with RF centers above or below the corresponding stimuli. We think it can be seen from the figure that the conclusion would not differ if we were to adjust these category boundaries to define “in” as inside the estimated circumference of the gabor patch stimuli.

Completing the requested analysis as described in point #2, a statistically significant relationship between spike counts and variation in eye position is observed in ~9% of units labeled “mixtures” (9/102), and ~5% of units not labeled mixtures (10/206). These proportions do not differ significantly (chi square test, p=0.17).

Altogether, we conclude that the correlation with eye position is insufficient to account for the bimodality in ~91% of the cases we labeled as “mixtures”. This is based on the following:

a)Correlation with eye position is rare (9%) and only somewhat more prevalent among “mixtures” vs.non mixtures (5%)b)Mixtures-with-eye-position-correlation appear quite randomly distributed in space in relationship to RF center and stimulus location.

Methodological details: Note that there were no included units with RF centers to the right of the rightmost stimulus, because it was usually placed on or near the vertical meridian. Note also that some units were excluded from this analysis due to a failure to identify an RF center with high confidence for that electrode location. Finally, since the electrode arrays were fixed in position, RF centers did not move much on a day to day basis. Accordingly, for this analysis we used one estimate of the RF centers for each monkey and applied it across all sessions for that monkey.

Changes to manuscript: The analysis is described in the methods “Correlations between firing rates and scatter in fixation position were assessed for the dual stimulus trials using the component of eye position that lay along a line connecting the two stimulus locations chosen for the recording session (see Figure 1 – Supplementary Figure 3 for results). “ and text has been added to the second paragraph of the Results: “Finally, we assessed the responses of individual units to ascertain what proportion of units showed a correlation between firing rate and fixational scatter; this proportion was small overall (4-9%) and did not co-vary with the outcomes of the main analyses of the study (see Figure 1 – Supplementary Figure 3 for details) “

The larger pink and green bull’s eye circles indicate the conditions showing a correlation with eye position (p<0.01) among the mixtures (pink) and non mixtures (green).

3. In addition, I think the authors should at least show the typical RF radius of the units (make a circle of radius of 0.5 degrees on a few of the units, perhaps the ones shown in Figure 4). We think it is important to show that the RFs do not encompass both stimuli, which is not clear from the plots.4. The fact that one mode of the bimodal distribution is zero in 91/100 cases should also be made clearer, perhaps in the methods section. Essentially the bimodal response in most cases is not A versus B, but A/B versus zero.

We agree.

Changes:

In the discussion, we have added the underlined sentence to paragraph 4 (with a little re-writing around it):

“It is interesting to note that we observed evidence of multiplexing each stimulus even in V1 where receptive fields are small and the stimuli we used did not themselves typically span more than one receptive field. Put another way, for most of these V1 “mixtures”, the observed fluctuations involved responding vs not responding rather than fluctuating between two different levels of responding. Thus the coarseness of tuning did not necessarily pose a problem for the encoding of these particular stimuli in this particular brain area, and yet fluctuations were observed. Thus, the precision of V1’s spatial code may not be the limiting factor. Multiplexing is likely to have some as yet unknown characteristic spatial scale that may be determined by the coarsest tuning evident at any stage in the sensory pathway. Future work in which stimuli are systematically varied to manipulate the amount of overlap in the activity patterns evoked in different brain areas by each stimulus alone are needed to answer this question. “

The numeric results are now presented in the legend to Figure 1 Supplementary Figure 3: “Finally, about 9% of “mixtures” (n=9) were responsive to both stimuli (i.e. RF centers were intermediate between the two stimuli and responses exceeded baseline firing by at least one standard deviation for both stimuli alone); among these 9 only 1 showed sensitivity to eye position (11%).”.

And we have added a sample RF size (0.5 degrees radius) to Figure 1 Supplementary Figure 1.